# The genomic and epigenomic evolutionary history of papillary renal cell carcinomas

Bin Zhu[1,15], Maria Luana Poeta[2,15], Manuela Costantini[2,3,15], Tongwu Zhang[1,15], Jianxin Shi[1], Steno Sentinelli[4], Wei Zhao[1], Vincenzo Pompeo[3], Maurizio Cardelli[5], Boian S. Alexandrov[6], Burcak Otlu[7], Xing Hua[1], Kristine Jones[8], Seth Brodie[8], Malgorzata Ewa Dabrowska[4,9], Jorge R. Toro[10], Meredith Yeager[8], Mingyi Wang[8], Belynda Hicks[8], Ludmil B. Alexandrov[7], Kevin M. Brown[1], David C. Wedge[11,12,13✉], Stephen Chanock[1,16], Vito Michele Fazio[9,14,16], Michele Gallucci[3,16] & Maria Teresa Landi[1,16✉]

Intratumor heterogeneity (ITH) and tumor evolution have been well described for clear cell renal cell carcinomas (ccRCC), but they are less studied for other kidney cancer subtypes. Here we investigate ITH and clonal evolution of papillary renal cell carcinoma (pRCC) and rarer kidney cancer subtypes, integrating whole-genome sequencing and DNA methylation data. In 29 tumors, up to 10 samples from the center to the periphery of each tumor, and metastatic samples in 2 cases, enable phylogenetic analysis of spatial features of clonal expansion, which shows congruent patterns of genomic and epigenomic evolution. In contrast to previous studies of ccRCC, in pRCC, driver gene mutations and most arm-level somatic copy number alterations (SCNAs) are clonal. These findings suggest that a single biopsy would be sufficient to identify the important genetic drivers and that targeting large-scale SCNAs may improve pRCC treatment, which is currently poor. While type 1 pRCC displays near absence of structural variants (SVs), the more aggressive type 2 pRCC and the rarer subtypes have numerous SVs, which should be pursued for prognostic significance.

[1] Division of Cancer Epidemiology and Genetics, National Cancer Institute, NIH, DHHS, Bethesda, MD 20892, USA. [2] Department of Bioscience, Biotechnology and Biopharmaceutics, University of Bari, 70126 Bari, Italy. [3] Department of Urology, "Regina Elena" National Cancer Institute, 00144 Rome, Italy. [4] Department of Pathology, "Regina Elena" National Cancer Institute, 00144 Rome, Italy. [5] Advanced Technology Center for Aging Research, IRCCS INRCA, 60121 Ancona, Italy. [6] Theoretical Division, Los Alamos National Laboratory, Los Alamos, NM 87545, USA. [7] Department of Cellular and Molecular Medicine and Department of Bioengineering and Moores Cancer Center, University of California, San Diego, La Jolla, CA 92093, USA. [8] Cancer Genomics Research Laboratory (CGR), Frederick National Laboratory for Cancer Research, Frederick, MD, USA. [9] Laboratory of Molecular Medicine and Biotechnology, University Campus Bio-Medico of Rome, 00128 Rome, Italy. [10] Washington, DC Veteran Affairs Medical Center, Washington, DC 20422, USA. [11] Big Data Institute, Old Road Campus, Oxford OX3 7LF, UK. [12] Oxford NIHR Biomedical Research Centre, Oxford OX4 2PG, UK. [13] Manchester Cancer Research Centre, Manchester M20 4GJ, UK. [14] Laboratory of Oncology, IRCCS H. "Casa Sollievo della Sofferenza", 71013 San Giovanni Rotondo, FG, Italy. [15]These authors contributed equally: Bin Zhu, Maria Luana Poeta, Manuela Costantini, Tongwu Zhang. [16]These authors jointly supervised this work: Stephen Chanock, Vito Michele Fazio, Michele Gallucci, Maria Teresa Landi. ✉email: david.wedge@manchester.ac.uk; landim@mail.nih.gov

Kidney cancer includes distinct subtypes[1] based on the presence of cytoplasmic (e.g., clear cell renal cell carcinoma, ccRCC), architectural (e.g., papillary renal cell carcinoma, pRCC), or mesenchymal (e.g., renal fibrosarcomas, rSRC) features. Rarer subtypes have also been defined by anatomic location (e.g., collecting duct renal cell carcinoma, cdRCC). Each of these subtypes has distinct implications for clinical prognosis. Within subtypes, there can be further differences in both tumor characteristics and prognoses. For example, papillary RCC are traditionally distinct into 2 types: (a) Type 1 with papillae covered by smaller cells with scant amphophilic cytoplasm and single cell layer, and (b) Type 2 with large tumor cells, often with high nuclear grade, eosinophilic cytoplasm and nuclear pseudostratification[2–4]. pRCC type 1 is more benign compared to the aggressive pRCC type 2. Recent cancer genomic characterization studies have revealed that the genomic landscape of major kidney cancer subtypes (e.g., ccRCC, pRCC, and chromophobe RCC) can be complex and differ substantially by subtype[5–7]. Patterns of intratumor heterogeneity (ITH) and tumor evolution have become the focus of intense investigation, primarily through multi-region whole-exome or whole-genome sequencing studies in ccRCC[8–10]. However, our understanding of the importance of ITH in other kidney cancer subtypes is either limited, such as for pRCC, the second most common kidney cancer subtype, where only four tumors have been characterized by whole-exome sequencing[11] or completely lacking, such as for cdRCC and rSRC. Moreover, previous ITH studies predominantly focused on single nucleotide variants (SNVs); little is known of the stepwise process in which additional genomic and epigenomic alterations (e.g., structural variants (SVs) or methylation changes) are acquired.

Herein, we fully characterize the whole genome and DNA methylation of pRCC and rarer kidney cancer subtypes, specifically examining both the core and periphery of selected tumors and, when available, metastatic lesions in order to investigate ITH and clonal evolution. We observe major differences from the previously studied clear cell renal cell carcinoma subtype. Specifically, pRCCs are characterized by clonal driver SNVs and arm-level somatic copy number alterations (SCNAs); modest intratumor heterogeneity of non-driver SNVs and methylation; and highly subclonal small SCNAs and SVs. Between pRCC subtypes, pRCC type 1 displays near absence of SVs, while pRCC type 2 and rare subtypes, which are more aggressive, have many SVs. Finally, integrated analysis of epigenomic and genomic data shows congruent patterns of evolution.

## Results

**Study design**. We conducted an integrative genomic and epigenomic ITH analysis of pRCC and rarer kidney cancer subtypes, each of which is distinct from the more commonly occurring ccRCC[12], and provide new insights into the clonal evolution of these subtypes. We examined multiple adjacent samples from the center of the tumor to the tumor's periphery as well as a normal sample ~5 cm distant from each tumor, and, when feasible, metastatic regions in the adrenal gland (Fig. 1a, "Methods" section). We performed 60X multi-region whole-genome sequencing (WGS, Supplementary Data 1) on 124 primary tumor and metastatic samples from 29 treatment-naive kidney cancers (Supplementary Table 1), as well as genome-wide methylation and SNP array profiling and deep targeted sequencing (average 500X coverage) (Supplementary Data 2) of 254 known cancer driver genes[13] (Supplementary Data 3). Tumors sequenced included 13 pRCC type 1 (pRCC1) tumors, 12 pRCC type 2 (pRCC2) tumors, and rarer subtypes (one each of cdRCC, rSRC, mixed pRCC1/pRCC2 and pRCC2/cdRCC) (Fig. 1b, "Methods" section). A section of each sampled region was histologically

examined: tumor samples included in the analyses had to exceed 70% tumor nuclei by pathologic assessment by a senior pathologist and the normal samples had no evidence of tumor nuclei. We also estimated the sample purity based on SCNAs or, in copy neutral samples, based on variant allele fraction (VAF) of single nucleotide variants (SNVs, Supplementary Fig. 1). The estimated purity based on WGS data were used to calculate precise cancer cell fractions (CCF) and hence to construct phylogenetic trees. Data on genome-wide methylation levels provided further information on epigenomic ITH.

**Frequency of somatic mutations and germline variants**. The average SNV and indel rates across tumors were 1.21/Mb and 0.18/Mb, respectively: on average, 1.00/Mb and 0.18/Mb for pRCC1; 1.46/Mb and 0.21/Mb for pRCC2. The SNV but not indel rates in pRCC2 were significantly higher than in pRCC1 (Wilcoxon test P-value = 0.03 for SNVs and P value = 0.65 for indels). For one tumor each of cdRCC, rSRC, mixed pRCC1/pRCC2 and pRCC2/cdRCC types, the SNV rates were, 1.46/Mb, 0.54/Mb, 0.95/Mb and 1.43/Mb, respectively; and the indel rates were 0.20/Mb, 0.05/Mb, 0.18/Mb and 0.13/Mb, respectively (Fig. 1c). Among the published kidney cancer driver genes, we observed that almost all driver SNVs (definition of driver mutations in "Methods" section) were clonal, in contrast to ccRCC[14]. Although we had only a single sample from 10 pRCC1 tumors, we conducted targeted sequencing to improve our knowledge of cancer driver mutations in this rare cancer type. In pRCC1 tumors, we found two *ATM*, two *MET* (both in the tyrosine kinase domain), and one in each *IDH1*, *EP300*, *KMT2A*, *KMA2C* and *NFE2L2* driver mutations. In pRCC2 tumors, we observed a *SMARCB1* driver mutation in one pRCC2; *TERT* promoter in two pRCC2; *SETD2*, *PBRM1* and *NF2* in one pRCC2 tumor each. We also found clonal indels in *NF2* in two tumors (cdRCC and mixRCC), and *MET* (mixRCC), *SMARCB1* (pRCC1) and *ROS1* (pRCC2) indels in one tumor each. We found no mutations in *TP53*, mutated in a high proportion of cases across cancer types[15], and no mutations in the 5'UTR region of *TERT*, which has been reported as mutated in a sizeable fraction of ccRCC[10] (Fig. 1c and Supplementary Fig. 2 and Supplementary Data 4 and 5). It has been previously reported that ~22.6% of pRCC do not harbor detectable pathogenic changes in any driver genes[11]. In a TCGA analysis of pRCC[6], overall ~23% of pRCC had no driver events. Here, we found four pRCC1 (31%) and three pRCC2 (25%) tumors, that had no detected SNVs or indels in previously reported driver genes, even after deep targeted sequencing. In these tumors, SNVs in other genes or other genomic alterations yet to be defined are the likely driver events.

An analysis of the germline sequencing data provided evidence of rare, potentially deleterious, germline variants in known cancer susceptibility genes ("Methods" section). These include two different variants in *POLE* in two different tumors; two different variants in *CHEK2* in two different tumors; one variant in *BRIP1* and *PTCH1* both in a single tumor; and additional rare variants, one per tumor (e.g., *TP53*, *MET*, *EGFR*, among others, Supplementary Data 6). This is consistent with a report on the relatively high frequency of germline mutations in cancer susceptibility genes in non-clear cell renal cell carcinomas[16].

**Phylogenetic trees show limited intratumor heterogeneity**. To explore ITH and to understand the sequence of genomic changes, we first constructed phylogenetic trees based on subclone lineages for 14 tumors with at least three regional samples per tumor (Fig. 2, phylogenetic trees of other samples in Supplementary Fig. 3), which included three pRCC1, eight pRCC2, and single tumors from three rarer subtypes. We used a previously reported

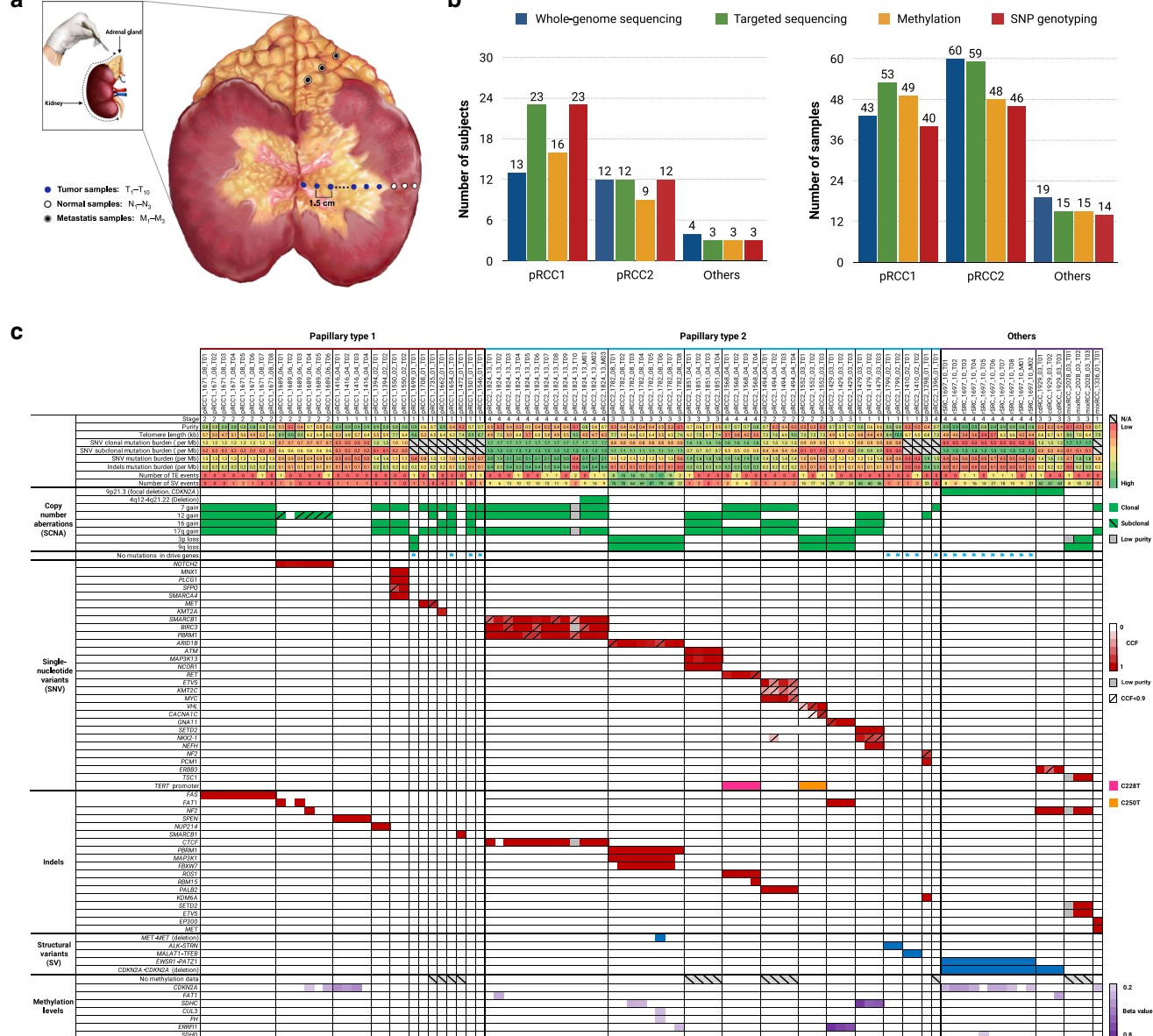

**Fig. 1 Study design and genomic landscape. a** A schematic illustration of the dissection of multiple tumor samples from the center of the tumor towards the tumor's periphery, plus metastatic samples in the adrenal gland as well as normal samples. For the analysis, the normal sample more distant from the tumor and with absence of tumor nuclei was chosen as reference. **b** Summary of subjects and samples that underwent different analyses based on DNA availability: whole-genome sequencing (124 samples from 29 subjects), deep targeted sequencing of cancer driver genes (139 samples from 38 subjects), genome-wide methylation (139 samples from 28 subjects) or SNP array profiling (only tumor samples, 101 samples from 38 subjects). **c** Tumor genomic alterations across histological subtypes. Shown are genome level changes, such as mutational burden, numbers of structural variants (SV) and retrotransposition events (TE), as well as other genomic alterations (denoted by different colors).

Bayesian Dirichlet process, DPClust[17], to define subclones based on clusters of SNVs sharing similar CCF, adjusting for SCNAs and purity estimated by the copy number caller Battenberg[18]. On average, we identified 5.3, 6.5 and 5.7 subclone lineages in pRCC1, pRCC2 and the rarer subtypes, respectively (Supplementary Data 7). We cannot exclude that, with deeper coverage across a larger number of SNVs and with more regions sampled from some of the tumors, DPClust could identify more subclones. Since ITH can be influenced by the number of samples sequenced per tumor, we used a recently proposed ITH metric, average pairwise ITH or *APITH*[19], to compare pRCC1 and pRCC2 ITH. APITH is defined as the average genomic distance across all pairs of samples per tumor and does not depend on the overall number of samples per tumor. We found that APITH of pRCC2

(mean = 26.66) is higher than APITH of pRCC1(mean = 16.20, unpaired student's *t* test *P* value = 0.03). We also investigated whether APITH was associated with tumor size, but found no association (*P* value = 0.38, all tumors; *P* value = 0.81, pRCC1; *P* value = 0.46, pRCC2).

Based on the identification of subclones, the SCHISM program[20] was applied to construct phylogenetic trees, which are consistent with the pigeonhole principle[18] and the 'crossing rule'[21]. The root of the phylogenetic tree represents germline cells without somatic SNVs; the knot between the trunk and branches is the most-recent common ancestor (MRCA), whose mutations are also shared by cells within all lineages. Phylogenetic trees with trunks that are long relative to the branches have lower levels of ITH. Each leaf represents a subclone; if a

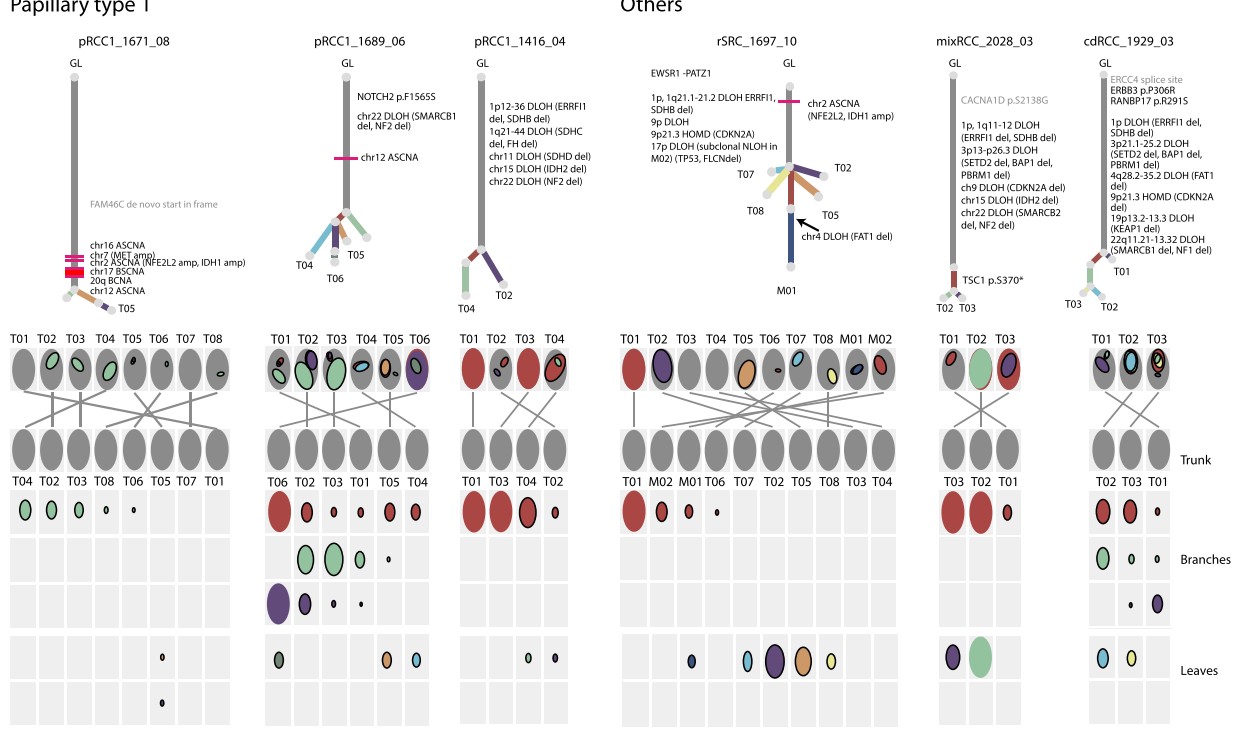

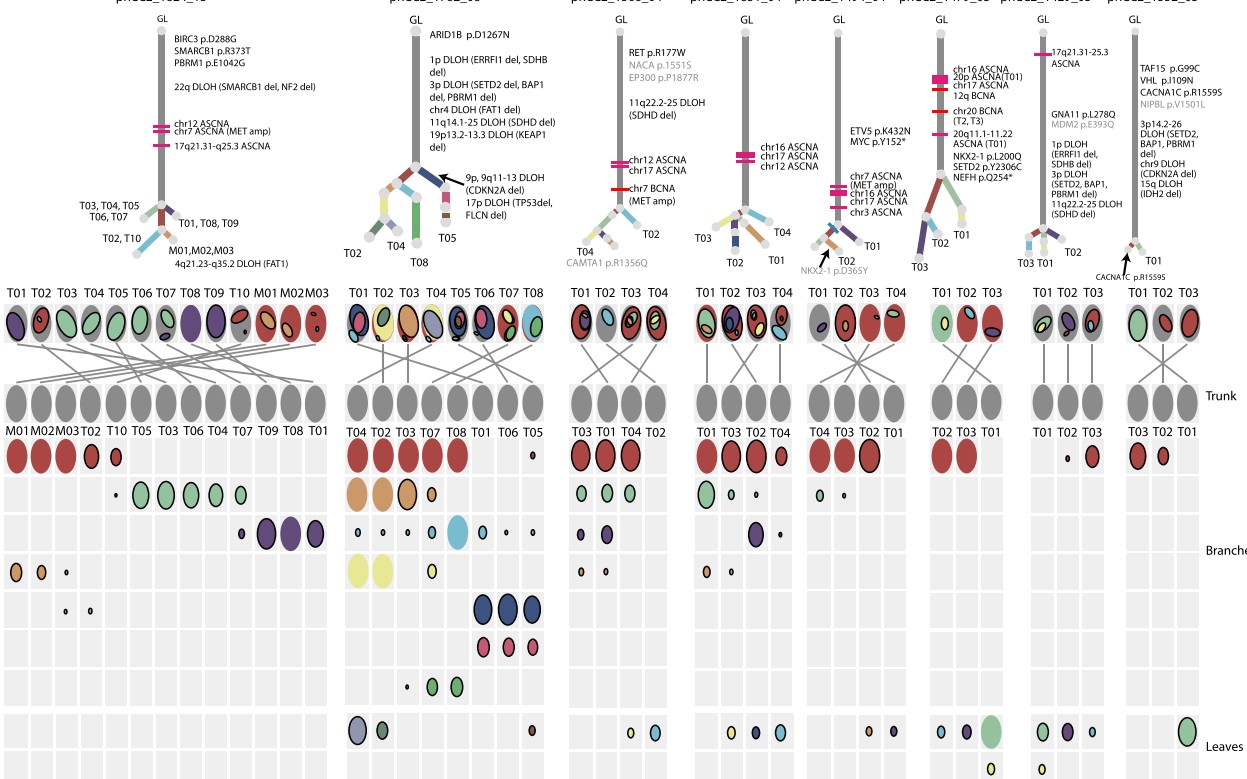

subclone exists in one region only, the leaf is annotated by the ID of this region. On average 70.0% of pRCC SNVs were in the trunk, with low ITH observed in both pRCC type 1 and 2 (Supplementary Fig. 4). This contrasts with previous findings in ccRCC[8,9,22] where approximately one-third of somatic mutations were truncal.

Segregating SNVs according to the genomic region in which they are located, we found a few pRCC tumors with higher ITH in promoters, 5′UTR and first exon regions (Supplementary Fig. 5). The metastatic samples in pRCC2_1824_13 (Figs. 1c and 2), which most likely originated in the primary tumor region T02 or T10, share the same driver mutations in *PBRM1* and *SMARCB*1.

**Fig. 2 Phylogenetic trees and oval plots for tumors with three or more samples.** Phylogenetic trees: the trees show the evolutionary relationships between subclones (annotated by different colors). Trunk and branch lengths are proportional to the number of substitutions in each clone cluster. Driver SNV and recurrent somatic copy number alterations are annotated on the trees. Tumor regions containing sample-specific subclones are indicated on the tree leaves. Oval plots: In the top rows the ovals are ordered based on the physical sampling of the tumor regions. Ovals are nested if required by the pigeonhole principle. The first row of the plot with nested ovals is linked by lines to the ovals ordered by the phylogenetic analysis, indicating intermixing of subclones spread across 2 or more tumor regions. In the matrix, each main clone (without solid border) and subclone (with solid border) is represented as a color-coded oval. The size of the ovals is proportional to the CCF of the corresponding subclones. Each column represents a sample. Oval plots are separated into three parts: trunk (top, CCF = 1 in all samples), branch (middle present in >1 sample but not with CCF = 1 in all samples), and leaf (bottom, specific to a single sample). GL germline, amp amplification, DLOH hemizygous deletion loss of heterozygosity, HET diploid heterozygous, NLOH copy neutral loss of heterozygosity, HOMD homozygous deletion, ASCNA allele-specific copy number amplification, BCNA balanced copy number amplification.

We found that subclones were not always confined to spatially distinct regions in pRCC tumors. For example, the purple clone cluster in pRCC1_1689_06 (Fig. 2) is present in neighboring regions T01, T02 and T03 and in distant region T06. Similarly, the red clone cluster in RCC2_1824_13 (Fig. 2) is observed only in two regions of the primary tumor, T02 and T10, which are approximately 12 cm apart. This suggests that pRCC tumor cells within the primary tumor may be motile, with the ability to skip nearby regions and spread directly to physically distant regions. This phenomenon has been previously observed in breast[23] and prostate cancers[24] but not, to our knowledge, in RCC. Alternatively, tumors may have grown predominantly as a single expansion producing numerous intermixed sub-clones that are not subject to stringent selection, as it has been proposed in the "Big Bang" model[25].

Many tumors displayed extensive intermixing of subclones, evidenced by the occurrence of a clone cluster at subclonal proportions across multiple samples. An example, pRCC2_1568_04, harbored four different clone clusters, each present across multiple samples. In total, nine of the 14 cases with three or more samples (Fig. 2) displayed intermixing of subclones spread across 2 or more regions. Since each of our tumors was sampled at ~1.5 cm intervals, it is apparent that intermixing extends across large geographical regions. In both of our metastatic cases (stage 4 at diagnosis), pRCC2_1824_13 and rSRC_1697_10, intermixing of subclones has extended to metastatic sites, pointing to the occurrence of polyclonal seeding as previously observed in metastatic prostate cancer[26].

**Clonality of copy number alterations varies by size**. We analyzed SCNAs from WGS data by considering both total and minor copy numbers (Supplementary Data 8 and Supplementary Table 2). If the SCNAs were shared across regions of the same tumor they were considered clonal; otherwise subclonal. The clonal proportion of SCNAs for each tumor was calculated as the proportion of the genome with identical SCNAs across all regions. pRCC1 and, to a lesser extent, pRCC2 showed recurrent amplification of chromosomes 7 (which includes the *MET* gene), 17, 12, and 16 (Supplementary Fig. 6). Notably, chr.3p loss, which is highly recurrent (~90%) in ccRCC[14,27], was present in 3 (25%) pRCC2 and 1 (7.7%) pRCC1 (Fisher's exact test *P* value = $7.49 \times 10^{-8}$ and *P* value = $1.04 \times 10^{-12}$, respectively). Among the samples with chr.3p loss, only one had a translocation with chr. 5q gain, while in ccRCC this translocation was shown in 43% of the samples with chr.3p loss[28]. We observed no genome doubling. On average, 3.3 and 22.6% of the genome had subclonal SCNAs (Supplementary Data 9) in pRCC1 and pRCC2, respectively (Figs. 1c and 3a, and Supplementary Fig. 7), with very few region-specific SCNAs (e.g., 13q in pRCC2_1782_08, Fig. 3a). Copy number type information is shown in Supplementary Fig. 7. We have labelled the recurrent SCNAs on the phylogenetic trees. In addition, we estimated the CCF of SCNAs at each region and calculated the average CCF of SCNAs across the primary and (if available) metastatic regions. We validated arm-level SCNA findings using our SNP array data and confirmed the

concordance across platforms, including estimation of purity and ploidy, and the largely clonal nature of these alterations (Supplementary Fig. 8). Most arm-level SCNAs were clonal (Fig. 3a) as previously suggested[10]. In contrast, we observed numerous small scale SCNAs shared by a subset of regions or existing in one region only, indicating SCNAs may be generated through changing mutational processes, with small scale SCNAs occurring in the later evolutionary phase (Fig. 3a). Further, the size of intra-chromosomal SCNAs was larger for clonal than subclonal events across all tumors (P-value = $1.3 \times 10^{-2}$, Wilcoxon rank test). Notably, all six pRCC2 tumors for which a comparison was possible (pRCC2_1429_03, pRCC2_1479_03, pRCC2_1552_03, pRCC2_1568_04, pRCC2_1782_08, pRCC2_1799_02) displayed this trend, while the two tumors belonging to rarer subtypes (cdRCC_1972_03, rSRC_1697_10) did not (Fig. 3b).

Hierarchical clustering showed that samples from the same tumors tended to cluster together (Supplementary Fig. 9), suggesting a higher inter-tumor heterogeneity than ITH. Metastatic lesions shared most SCNAs with their primary tumors, but also displayed metastasis-specific SCNAs (e.g., hemizygous deletion loss of heterozygosity in 4q of pRCC2_1824_13, Fig. 3c), indicating ongoing SCNA clonal evolution during metastasis. Among the rarer subtypes, both rSRC and cdRCC had clonal focal homozygous deletions of *CDKN2A* at 9p21.3 (Fig. 1c and Supplementary Figs. 10 and 11, Supplementary Data 10).

We further ordered the occurrence of driver mutations relative to somatic copy number gains or loss of heterozygosity (LOH)[18,29] and were able to infer the timing of some driver mutations (Supplementary Data 11). For example, the SMARCB1 p.R373T mutation occurred earlier than the 22q LOH in pRCC2_1824_13_T08, and the truncated mutation KMT2C p.S789* occurred later than the chr7 amplification in pRCC2_1494.

**Frequency of SVs differs between pRCC1 and pRCC2**. Somatic SVs were called by the Meerkat algorithm[30], which distinguishes a range of SVs and plausible underlying mechanisms, including retrotransposition events. pRCC2 had significantly more SV events per tumor, averaging 23.6, as compared to 1.2 events per tumor in pRCC1 (*P* value = $1.07 \times 10^{-3}$, Wilcoxon rank test, Supplementary Data 12). Tandem duplications, chromosomal translocations, and deletions were the most prevalent types of variant (36.4, 34.0, and 29.4%, respectively, Fig. 4a). Some SVs involved known cancer driver genes (Fig. 1c), including a deletion within *MET* in one pRCC2, and several fusions involving genes previously reported in renal cancer or other tumors. These included *ALK/STRN*[31] and *MALAT1/TFEB*[32] in two different pRCC2 and *EWSR1/PATZ1*[33] in the rSRC. We had high quality RNA material to validate the latter two SVs (Supplementary Fig. 12). We note that one tumor (pRCC2-1410), which had the morphological features of pRCC2, showed the classic *MALAT1-TFEB* gene fusion. Thus, it should be considered a MiT family translocation renal cell carcinoma (TRCC)[32,34]. As expected for

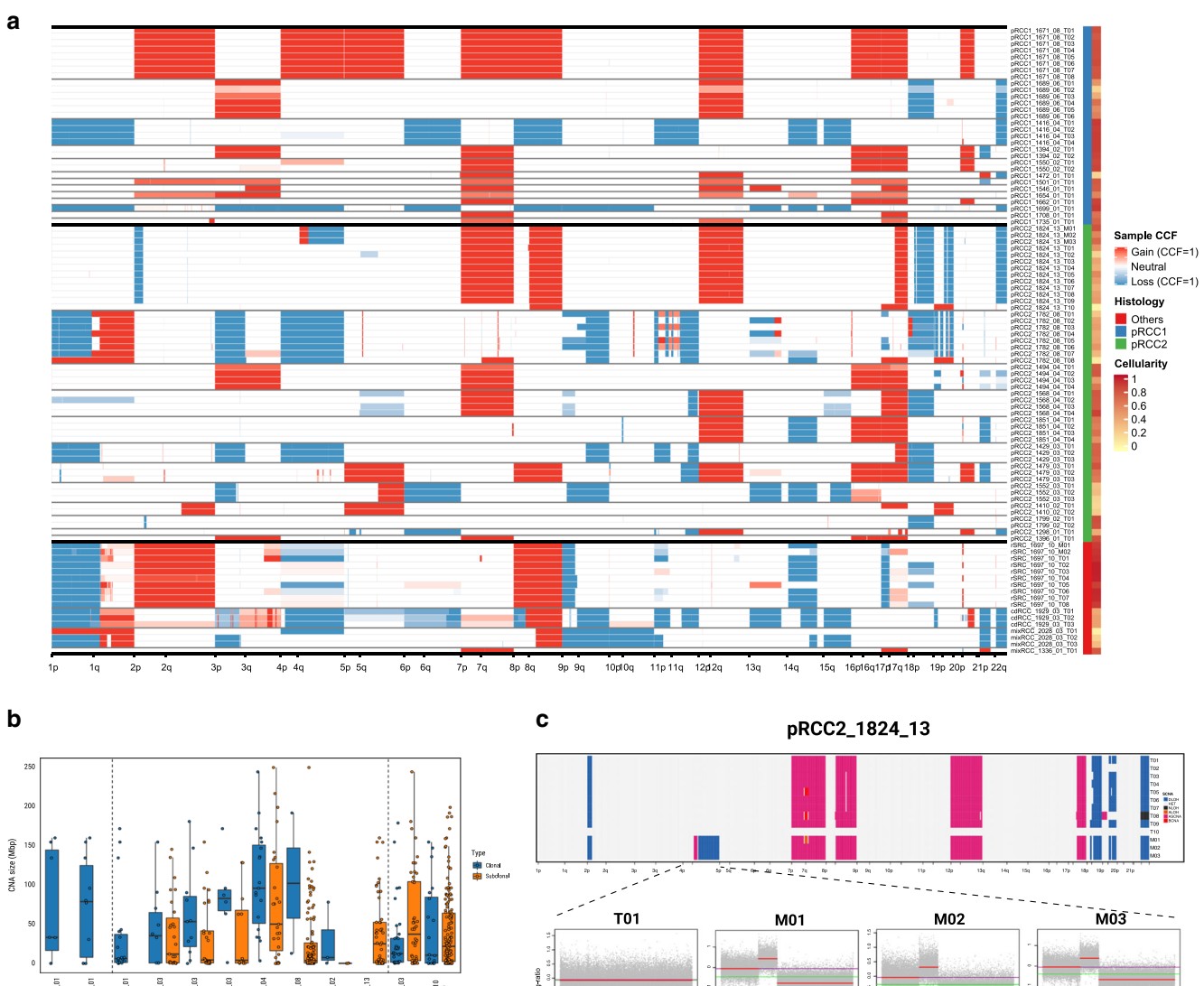

**Fig. 3 Somatic copy number alterations (SCNAs). a** Genome-wide sample cancer cell fraction (CCF) profiles across tumors. Samples are labeled by histology subgroup and cellularity (or called purity). (**b**) Size of SCNAs identified in each tumor, separated into clonal alterations (purple), which have a shared breakpoint across all samples from a tumor and subclonal alterations (yellow), which have breakpoints that are found only in a subset of samples from a tumor. Where clonal and subclonal SCNAs are both identified in a tumor, they are shown side by side. The numbers of clonal and subclonal SCNAs are included in the parentheses following the tumor ID, such as pRCC1_1472_01(n = 7,0). Vertical lines separate pRCC1 (left), pRCC2 (middle) and rare subtypes (right). In each box-and-whisker plot, the line dividing the box represents the median; the ends of the box are the lower (Q1) and upper (Q3) quartiles; the whiskers are extended to Q1-1.5xIQR and Q3 + 1.5IQR with IQR = Q3–Q1. Each circle represents a data point of SCNA size. (**c**) SCNAs of tumor pRCC2_1824_13. Top panel: genome-wide SCNAs on ten primary tumors (T01-T10) and three metastatic samples (M01, M02 and M03); T10 has low purity and has no SCNAs. Bottom panels: metastatic sample-specific SCNAs on chromosome 4 for total copy number log-ratio (red line: estimated total copy number log-ratio; green line: median; purple line: diploid state). DLOH: hemizygous deletion loss of heterozygosity; HET: diploid heterozygous; NLOH: copy neutral loss of heterozygosity; ALOH: amplified loss of heterozygosity; ASCNA: allele-specific copy number amplification; BCNA: balanced copy number amplification.

this subtype, this patient had a good prognosis (long survival and no metastasis).

Substantial variation in both the number and type of SVs was observed between tumors (Fig. 4a), again suggesting strong inter-tumor heterogeneity. Some tumors, particularly amongst the pRCC1s, had almost no SVs (e.g., pRCC1_1671_08 in Fig. 4b); some had SVs clustered in a hotspot (Supplementary Fig. 13), while still others had many SVs, like pRCC2_1824_13 (Fig. 4c) and pRCC2_1782_08 (Fig. 4d), the latter showing high genomic instability. Interestingly, pRCC2_1782_08 had a high number of LINE-1 clonal retrotransposition events detected by TraFiC[35]

(Fig. 4a and Supplementary Fig. 14), while somatic retro-transposition events were rarely detected in the remaining samples (Supplementary Data 13), as was observed in ccRCC and chromophobe RCC[36]. At least three transposon insertions could have potentially affected the expression of proteins involved in chromatin regulation and chromosome structural maintenance and, in turn, the maintenance of genome integrity in this tumor (Supplementary Method).

In contrast to arm-level SCNAs (Fig. 3a), most SVs were subclonal or late events within the tumors (Supplementary Fig. 15), appearing on the branches of the phylogenetic trees.

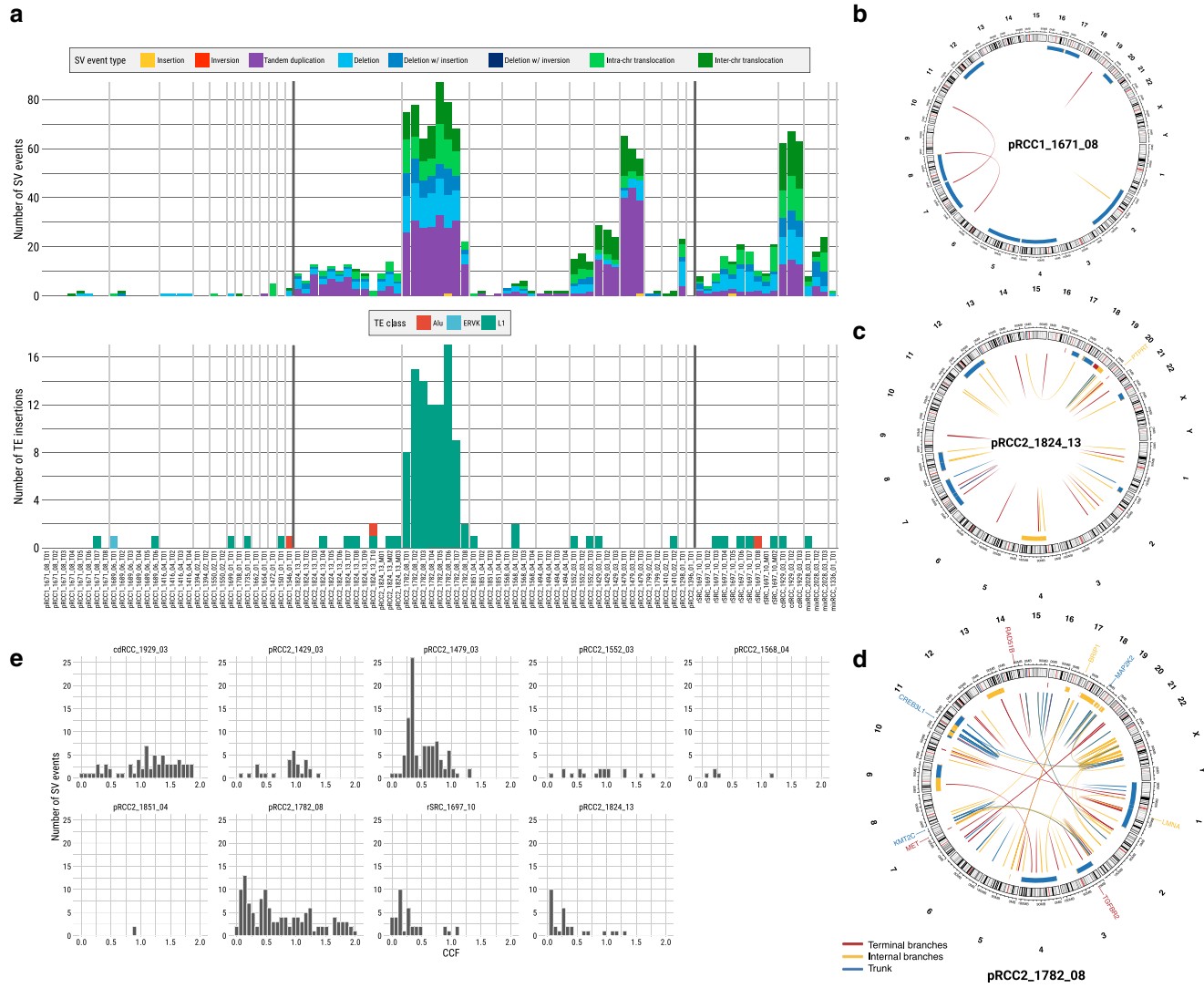

**Fig. 4 Structural variants (SV) and retrotransposition events (TE). a** Frequency of SV events and TE insertions for each sample. **b–d** Circos plots for SV events for three tumors; involved driver genes are noted. **e** The distribution of mean cancer cell fraction (CCF) of SVs across tumors. Alu elements originally characterized by the action of the Arthrobacter luteus (Alu) restriction endonuclease, ERVK mouse endogenous retrovirus K, L1 Long interspersed element-1.

Specifically, on average 40% of SVs were shared among all regions of a tumor. This is consistent with the average CCF of SVs across regions; in most of the tumors with more than three sampled regions, the average CCF was less than 0.75/tumor (Fig. 4e). We validated 88% of the WGS-Meerkat detected SV events and using a PCR-based sequencing methodology (Ampliseq; Supplementary Fig. 16 and Supplementary Method). It is notable that PCR sequencing also validated the clonal/subclonal status, defined by presence in all or just a subset of samples, of 83% of the SVs, and confirmed that SVs in pRCC have high ITH. Moreover, we compared the breakpoints between all SCNAs (estimated by Battenberg) and SVs (estimated by Meerkat). These results suggest that Battenberg (and probably copy number callers in general) has poor sensitivity for calling certain types of SVs and shows the value of combined analysis of SVs and SCNAs (Supplementary Fig. 17, details in Supplementary Method).

**Mutational signatures and telomere length**. De novo extraction of SNV mutational signatures identified the patterns of four distinct mutational signatures, termed signatures A through D (Supplementary Fig. 18). Comparison of these four de novo

deciphered signatures to the global consensus set of mutational signatures[37] revealed that signatures A through D are linear combinations of six previously known SNV mutational signatures (Supplementary Table 3): single base signatures (SBS) 1, 2, 5, 8, 13, and 40. Signatures 5 and 40 (cosine similarity: 0.83) are both of unknown etiology and were found across all examined RCC subtypes (mean contributions 32.6% and 59.9%, respectively, Supplementary Data 14). We also observed a small proportion of mutations attributed to the clock-like[38] mutational signature 1 (3.5% of total SNVs) and signature 8 (1.4%), which has unknown etiology. Moreover, we found that the numbers of clonal mutations assigned to signature 1, 5 or 40 were significantly associated with age at diagnosis (Supplementary Fig. 19a, SBS1 vs age: Pearson's correlation coefficient ($R$) = 0.46, P-value = 0.013; SBS5 vs age: $R$ = 0.40; $P$ = 0.033; SBS40 vs age: $R$ = 0.48, $P$ = 0.009), while the number of subclonal mutations assigned to signature 1, 5 or 40 was not (Supplementary Fig. 19b). Further, low mutational activity was detected for signature 2 (0.6%) and signature 13 (0.7%), both attributed to the activity of the APOBEC family of deaminases (Supplementary Fig. 20). All signatures were found in both clonal and subclonal SNVs (Supplementary Fig. 21) and

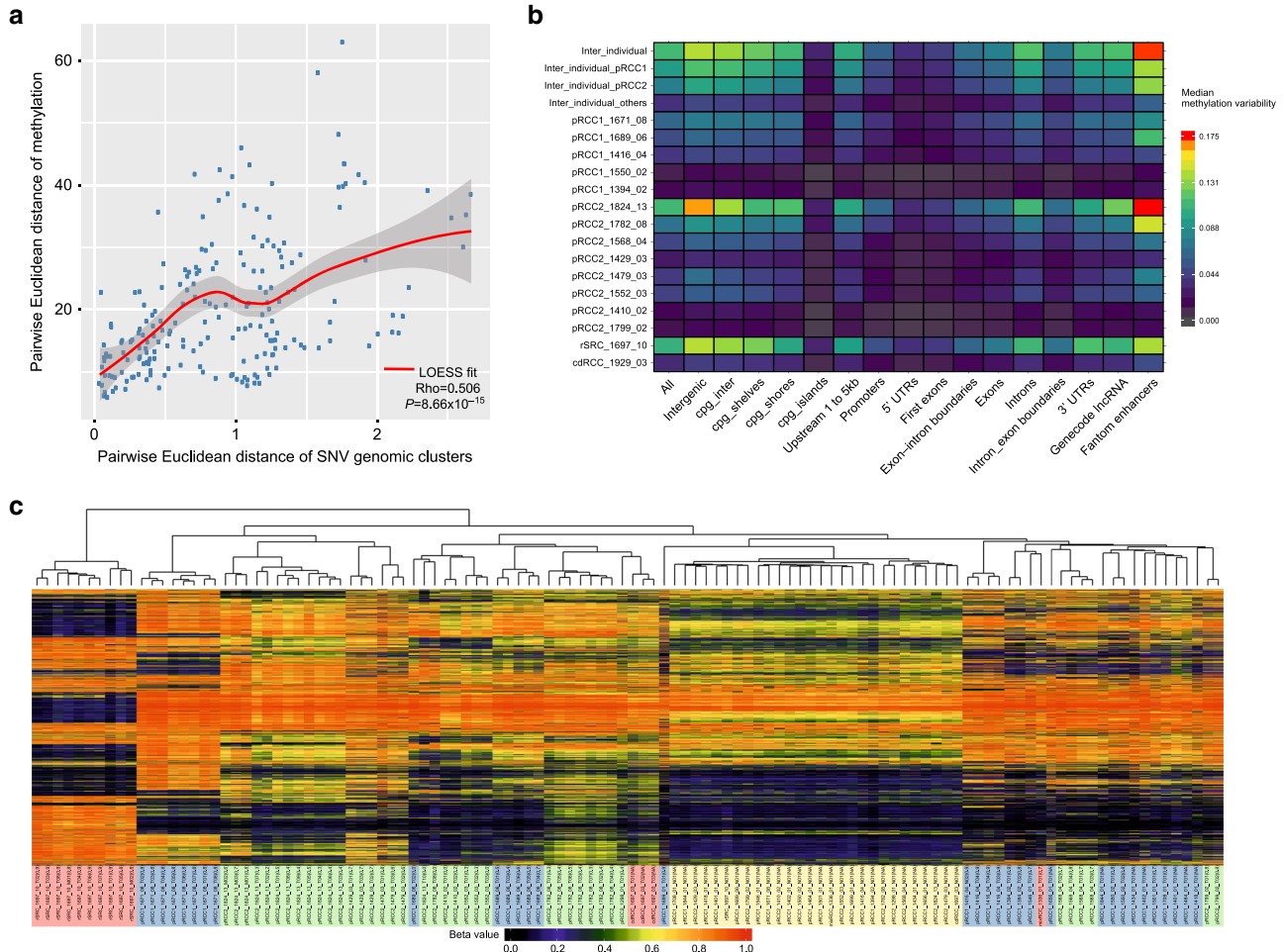

**Fig. 5 Intratumor heterogeneity (ITH) of methylation profiles. a** Scatter plots of pairwise distance between methylation and single nucleotide variant clusters. LOESS (Locally Weighted Scatterplot Smoothing) fitted curve is shown in red line with 95% confident interval in gray shaded area. Spearman's rank correlation rho is shown on the bottom right. For the two-sided test of rho = 0, the test statistic S is 719790 based on the algorithm AS 89. The exact *P* value is $8.66 \times 10^{-15}$. **b** Methylation ITH on genomic regions for each sample and tumor subtype. **c** Unsupervised hierarchical clustering of methylation profiles measured by the methylation level as beta value for the top 1% most variable methylation probes. Sample IDs are followed by the purity estimated by SCNAs or SNV VAF in parentheses. The background colors of the sample IDs represent different histological subtypes and tumor or normal tissue samples.

varied only slightly between primary and metastatic samples (Supplementary Fig. 22). Additional characteristics of SNV and Indel mutational signatures are included in the Supplementary Method.

We estimated telomere length (TL) based on the numbers of telomere sequence (TTAGGG/CCCTAA)$_4$ using TelSeq[39]. The normal and metastatic tissue samples on average had longer (8.51 kb, one side Mann–Whitney *U* test *P* value = $1.16 \times 10^{-6}$) and shorter (4.4 kb, *P* value = $1.96 \times 10^{-3}$) TL, respectively than the primary tumor tissue samples (6.12 kb) (Supplementary Fig. 23 and Supplementary Data 15).

**DNA Methylation ITH**. To analyze methylation ITH, we chose the 1% of methylation probes in CpG sites with the greatest intratumoral methylation range and calculated the methylation ITH based on the Euclidean distances between regions. In general, methylation ITH was not high and similar across histological subtypes (Kruskal–Wallis Test: *P* value = 0.675) (Supplementary Fig. 24). For most cases with four or more samples, we calculated the Euclidean distance separately for SNVs and methylation levels (using the top 5000 most variable CpG probes) for each pair of tumor samples within a tumor. We found that the difference in

methylation patterns between pairs of samples correlated strongly with pairwise differences in subclonal SNVs ($P < 0.0001$, $R = 0.5$) (Fig. 5a), implying congruence between genomic and epigenomic evolutionary histories. Although methylation ITH was generally low, the analysis showed greater ITH in enhancer regions, and no ITH in promoter/5'UTR/1st exons or CpG island regions (Fig. 5b), suggesting a possible role of methylation ITH in shaping regulatory function, but tight control of the genome regions directly affecting gene expression.

Unsupervised clustering analysis based on the 1% most variable methylation probes clearly separated tumor samples from normal samples, and pRCC tumors from renal sarcoma (Fig. 5c). Moreover, samples with purity <30% clustered together but separately from the normal or the tumor tissue samples, likely because they were enriched with stromal, immune or other non-epithelial cells. Similarly, although metastasis samples in pRCC2_1824_13 appear to arise from the T02 and T10 regions based on the phylogenetic analysis (Fig. 2), they cluster separately from any tumor region likely because methylation reflects the different tissue type (adrenal gland). This finding is comparable to what has been reported in the TCGA pan-can analyses, where methylation profiles have been used to infer cell-of-origin patterns across cancer types[40]. Future studies

should evaluate other epigenetic modifications to provide more comprehensive details of epigenomic evolutionary history of pRCC.

## Discussion

Multi-region whole-genome sequencing demonstrates that papillary renal cell carcinomas and rarer renal cancer subtypes generally have much less driver gene mutation and copy number alteration intra-tumor heterogeneity than clear cell renal cell carcinomas. In pRCCs, evolution of the epigenome occurs in step with genomic evolution, although DNA methylation ITH in promoter regions was lower suggesting a tighter regulation of the somatic epigenome.

Large-scale copy number aberrations, often associated with inter-chromosomal translocations, were frequently clonal across all samples from a tumor. The observed clonal status of SCNAs may be the result of an early burst of large-scale genomic alterations, providing growth advantage to an initiating clone that then expands stably. At the time of diagnosis, the descendants of these cells, which have accumulated additional genetic aberrations, appear to be characterized by a single or small number of large SCNA events. In support of this hypothesis, bulk- and single-cell based copy number and sequencing studies of breast and prostate cancers[41–43] have suggested that complex aneuploid copy number changes may occur in only a few cell divisions at the earliest stages of tumor progression, leading to punctuated evolution.

The ITH of SNVs was greater than that of large SCNAs, and ITH of small SVs was even greater. The few SNVs, indels and fusions we identified in known cancer driver genes were clonal in all samples, from both pRCC subtypes. Thus, our data indicate that papillary renal cell carcinomas initiate through a combination of large clonal SCNAs and mutations in different driver genes, while tumor progression is further promoted by additional SNVs, small scale SCNAs and SVs.

The mechanisms of SVs formation are largely unknown. A landscape description of breast cancer[44] and a recent structural variant analysis in PCAWG[45] identified different signatures of structural variants, separated by size. Taken all together, these findings suggest that there are different mutational and repair processes operating at different scales and future research should be directed towards further elucidating the causal mechanisms.

Although ITH is generally correlated with the number of samples per tumor, the increase in ITH in the order (large SCNAs – SNVs - small SVs) was consistent across both pRCC subtypes and irrespective of the number of tumor samples. Moreover, we used an estimate of ITH that is not affected by the number of samples sequenced per tumor (APITH)[19] and found that APITH in pRCC2 was significantly higher than ITH in pRCC1. ITH has been found to impact prognosis or response to treatment across cancer types[46,47], highlighting the importance of further exploring pRCC ITH in light of a possible treatment strategy.

Signatures SBS5 and SBS40 accounted for 92.5% of all somatic mutations observed in pRCC. High frequency of signature SBS40 has been found in kidney cancer in previous studies, possibly due to the organ's cells constant contact with mutagens during the blood filtration process[37]. Both signatures have unknown etiology, but they have been associated with age at diagnosis across most human cancers[37]. These "flat" signatures are correlated to each other and likely harbour common mutation components. In our study, clonal mutations attributed to signatures 1, 5, and 40 were all significantly correlated with age of diagnosis, suggesting that they may be the result of a lifetime accumulation of mutations. Future experimental studies are necessary to investigate the molecular and mutational underpinning of signatures 5 and 40. Notably, among the 29 subjects with WGS data, 13 were never smokers, 4 current smokers, 6 former smokers and 6 had unknown smoking status. However, we found no tobacco smoking signature SBS4, as previously observed in kidney and bladder cancers[48].

In our analysis of a series of samples from the tumor center to the tumor periphery at precise distance intervals, we found that tumors are not necessarily composed of separate subclones in distinct regions of a tumor. Instead, we observed widespread intermixing of subclonal populations. In our 2 metastatic cases, the subclones remained mixed when spread to distant sites, possibly indicating polyclonal seeding of metastases[26]. Evidence for tumor cells transiting large distances across the primary tissue was also seen in four cases (Fig. 2).

In addition to provide insight into the natural history of these tumors, understanding the clonal expansion dynamics of these cancers has potentially important implications for diagnosis and treatment. Although based on a limited number of tumors, the observed clonal patterns of both large scale SCNAs and SNVs/indels in driver genes suggest a single tumor biopsy would be sufficient to characterize these changes. However, although targeted therapies against the few driver gene mutations or rare germline variants we identified (e.g., *MET, VHL, PBMR1, ARID1B, SMARCA4, ALK, TFEB*) are either available or presently being evaluated in clinical trials, therapies against SCNAs are critically needed. Compounds that inhibit the proliferation of aneuploid cell lines[49] or impact the more global stresses associated with aneuploidy in cancer or target the bystander genes that are deleted together with tumor suppressor genes (collateral lethality)[50–52] are encouraging and should be further explored. Further therapeutic challenges for the renal cell tumors we studied are provided by the subclonal nature of SVs as well as the low mutation burden and the notable lack of *TP53* mutations, both of which may hinder response to immune checkpoint inhibitors[53–55]. Notably, while the numbers of SCNAs were similar between pRCC1 and pRCC2, the number of SV events, and – to a lesser extent – the SNV events were higher in pRCC2 in parallel with the more aggressive tumor behavior of this subtype. These findings emphasize the importance of further investigating these changes for prognostic significance in future larger studies.

## Methods

**Patients and specimens**. This study was based on archived samples collected at the Regina Elena Cancer Institute, Rome, Italy. Written informed consent to allow banking of biospecimens for future scientific research was obtained from each subject. This work was excluded from the NCI IRB Review per 45 CFR 46 and NIH policy for the use of specimens/data by the Office of Human Subjects Research Protections (OHSRP) of the National Institutes of Health. The data were anonymized.

The study population comprise 39 patients with kidney cancers, including 23 with papillary type 1 (pRCC1); 12 with papillary type 2 (pRCC2); and one each with collecting duct tumor (cdRCC); renal fibrosarcoma rSRC (with negative stain for AE1/AE3, PAX8, CD99, FLI-1, WT1, actine ml, desmine, Myod-1, and HMB45; and positive staining for vimentine and S-100 (focal)); mixed pRCC1/pRCC2 and an unclassified renal cancer with mixed features of pRCC2 and cdRCC (mixRCC). The histological diagnosis was reviewed by an expert uropathologist (S.S.) based on the 2016 World Health Organization (WHO) classification of renal tumors[1]. Although our pathologist reviewed all available tissue blocks from each tumor, we cannot exclude the possibility that some of these tumors have mixed histologies (e.g., papillary types 1 and 2) in sections that were not available for histological review. Moreover, the distinction between pRCC1 and pRCC2 can be sometimes murky because of overlapping features and no immunohistochemistry or molecular marker can conclusively distinguish the two subtypes. For example, trisomies 7 and 17 are frequent in pRCC1 but can be also found, less frequently, in pRCC2[4]. There could also be tumors with one dominant histology and a small component of a different histology. For example, pRCC2_1552_03 was a pRCC2 with small areas with clear cells, which may explain the VHL mutation we identified in this tumor. Histological images of all tumors can be found in the Supplementary Histological Images file (Supplementary Fig. 25).

Based on DNA sample availability, we conducted whole-genome sequencing (WGS) on 124 samples from 29 subjects, deep targeted sequencing on 139 samples from 38 subjects, SNP array genotyping on 101 samples from 38 subjects, and

genome-wide methylation profiling on 139 samples from 28 subjects (Fig. 1b, more details in Supplementary Fig. 26). All assays were performed on tumor, metastasis and normal tissue samples, with the exception of the SNP array genotyping, which was conducted only on tumor samples.

**Study Design**. All tumors were treatment-naive. We used a study design with multiple tumor samples taken at a distance of ~1.5 cm from each other starting from the center of the tumor towards the periphery, plus multiple samples from the most proximal to most distant area outside the tumor. When present, we also collected multiple samples from metastatic regions outside the kidney (adrenal gland) (Fig. 1a). For the analyses presented here, we analyzed all multiple tumor and metastatic samples/tumor with at least 70% tumor nuclei at histological examination. As a reference, we used the furthest "normal" sample from each tumor, with histologically-confirmed absence of tumor nuclei.

**Whole-genome sequencing**. Genomic DNA was extracted from fresh frozen tissue using the QIAmp DNA mini kit (Qiagen) according to the manufacturer's instructions. Libraries were constructed and sequenced on the Illumina HiSeqX at the Broad Institute, Cambridge, MA with the use of 151-bp paired-end reads for whole-genome sequencing (mean depth = 65.7× and 40.1×, for tumor and normal tissue, respectively). Output from Illumina software was processed by the Picard data-processing pipeline to yield BAM files containing well-calibrated, aligned reads to genome-build hg19. All sample information tracking was performed by automated LIMS messaging. More details are included in the Supplementary Method.

**Genome-wide SNP genotyping**. Genome-wide SNP genotyping, using Infinium HumanOmniExpress-24-v1-1-a BeadChip technology (Illumina Inc. San Diego, CA), was performed at the Cancer Genomics Research Laboratory (CGR). Genotyping was performed according to manufacturer's guidelines using the Infinium HD Assay automated protocol. More details are included in the Supplementary Method.

**Targeted Sequencing**. A targeted driver gene panel was designed for 254 candidate cancer driver genes[13]. For each sample, 50 ng genomic DNA was purified using AgencourtAMPure XP Reagent (Beckman Coulter Inc, Brea, CA, USA) according to manufacturer's protocol, prior to the preparation of an adapter-ligated library using the KAPA JyperPlus Kit (KAPA Biosystems, Wilmington, MA) according to KAPA-provided protocol. Libraries were pooled, and sequence capture was performed with NimbleGen'sSeqCap EZ Choice (custom design; Roche NimbleGen, Inc., Madison, WI, USA), according to the manufacturer's protocol. The resulting post-capture enriched multiplexed sequencing libraries were used in cluster formation on an Illumina cBOT (Illumina, San Diego, CA, USA) and paired-end sequencing was performed using an Illumina HiSeq 4000 following Illumina-provided protocols for 2 × 150 bp paired-end sequencing at The National Cancer Institute Cancer Genomics Research Laboratory (CGR). More details are included in the Supplementary Method.

**Methylation analysis**. A concentration of 400 ng of sample DNA, according to Quant-iTPicoGreen dsDNA quantitation (Life Technologies, Grand Island, NY), was treated with sodium bisulfite using the EZ-96 DNA Methylation MagPrep Kit (Zymo Research, Irvine, CA) according to manufacturer-provided protocol. Bisulfite conversion modifies non-methylated cytosines into uracil, leaving 5-methylcytosine (5mC) and 5-hydroxymethylcytosine (5hmC) unchanged. High-throughput epigenome-wide methylation analysis, using Infinium MethylationE-PICBeadChip (Illumina Inc., San Diego, CA) which uses both Infinium I and II assay chemistry technologies was performed according to manufacturer-provided protocol at CGR. More details are included in the Supplementary Method.

**Whole-Genome data processing and alignment**. The WGS FASTQ files were processed and aligned through an in-house computational analysis pipeline, according to GATK best practice for somatic short variant discovery (https://software.broadinstitute.org/gatk/best-practices/). First, quality of short insert paired-end reads was assessed by FASTQC (https://www.bioinformatics.babraham.ac.uk/projects/fastqc/). Next, paired-end reads were aligned to the reference human genome (build hg19) using BWA-MEM aligner in the default mode[56]. The initial BAM files were post-processed to obtain analysis-ready BAM files. In particular, sequencing library insert size and sequencing coverage metrics were assessed, and duplicates were marked using Picard tools (https://broadinstitute.github.io/picard/); indels were realigned and base quality scores were re-calibrated according to GATK best practice; In addition, BAM-matcher was used to determine whether two BAM files represent samples from the same tumor[57]; VerifyBamID was used to check whether the reads were contaminated as a mixture of two samples[58].

**Somatic mutation calling from whole-genome sequencing data**. The analysis-ready BAM files from tumor, metastasis, and matched normal samples were used to call somatic variants by MuTect2 (GATK 3.6, https://software.broadinstitute.org/gatk/documentation/tooldocs/current/org_broadinstitute_gatk_tools_walkers_

cancer_m2_MuTect2.php) with the default parameters. In the generated VCF files, somatic variants notated as "Somatic" and "PASS" were kept. A revised method described by Hao, et al.[59] was used to further filter the somatic variants. More details are included in the Supplementary Method. For indels, we reported those that overlapped across three different software, mutect2[60], strelka2[61], and tnscope[62]. Indels were left-aligned and normalized using bcftools. The intersection of "PASS" indels from all three calling tools were combined by GATK "CombineVariants". Additional filters were applied to the final set before downstream analysis: tumor alternative allele fraction >0.04; normal alternative allele fraction <0.02; tumor total read depth > = 8; normal total read depth > = 6; and tumor alternative allele read depth >3.

**Identification of putative driver mutations and driver genes**. To create putative cancer driver gene and mutation lists, we first listed the putative cancer driver genes on the basis of recent large-scale TCGA Pan-kidney cohort (KICH + KIRC + KIRP) sequencing data (http://firebrowse.org), i.e., the significantly mutated genes identified by MutSig2CV algorithm with q value less than 0.1. In addition, we included the genes from the COSMIC cancer gene census list (May 2017, http://cancer.sanger.ac.uk/census) in the putative kidney driver gene set. Putative driver mutations were defined if they met one of the following requirements: (i) if the variant was predicted to be deleterious, including stop-gain, frameshift and splicing mutation, and had a SIFT[63] score < 0.05 or a PolyPhen[64] score >0.995 or a CCAD[65] score >0.99; or (ii) If the variant was identified as a recurrent hotspot (statistically significant, http://cancerhotspots.org) or a 3D clustered hotspot (http://3dhotspots.org) in a population-scale cohort of tumor samples of various cancer types using a previously described methodology[66,67].

**Germline variants in cancer susceptibility genes**. A germline variant was included if its minor allele frequency was <0.1% in an Italian whole-exome sequencing data from 1,368 subjects with no cancer[68] and the GnomAD European-Non Finnish-specific data from 12,897 subjects[69].

**Mutational signature analysis from whole-genome sequencing data**. Mutational signatures were extracted using our previously developed computational framework SigProfiler[70]. A detailed description of the workflow of the framework can be found in Refs. [37,71], while the code can be downloaded freely from: https://www.mathworks.com/matlabcentral/fileexchange/38724-sigprofiler. Detailed description of the methodology can be found in Supplementary Method.

**Mutation clustering and phylogenetic tree construction and annotation**. Clustering of subclonal somatic substitutions in whole-genome data were analyzed using a Bayesian Dirichlet process (DP) in multiple dimensions across related samples as previously described[26].Copy number changes called by the Battenberg algorithm and read count information of each mutation across all regions in the same tumor were used to calculate cancer cell fraction (CCF) and prepared as input for DPClust. Clone clusters were identified as local peaks in the posterior mutation density obtained from the DP. For each cluster, a region representing a 'basin of attraction' was defined by a set of planes running through the point of minimum density between each pair of cluster positions. Mutation were assigned to the cluster in whose basin of attraction they were most likely to fall, using posterior probabilities from the DP. This process was extended into multiple dimensions for the patients with multiple related samples. The following criteria were applied to remove the clusters: 1) cluster included less than 1% total mutations; 2) most mutations in cluster were localized to a small number of chromosomes; 3) conflicting cluster due to two principles as previously described[72]: pigeonhole principle and crossing rule.

The tumor subclonality phylogenetic reconstruction algorithm SCHISM[20] (SubClonal Hierarchy Inference from Somatic Mutations) was used to infer phylogenetic trees based on the CCF of final clone clusters. The phylogenetic tree and clone cluster relationship were manually created and organized according to previous publication[26]. The mutations and/or copy number alterations in potential driver genes as well as the recurrent copy number alterations were marked on the trees. Palimpsest[29] was used to time the chromosomal duplications. The ratio of duplicated/non-duplicated clone mutations were used to time these events, with early events having a low amount of duplicated mutations as compared to late events[18,73]. The relative order of these duplication events was then mapped on the trunk of the trees.

**Somatic copy-number alteration (SCNA) analysis**. Identification of clonal and subclonalcopy number alterations for each sample was performed with the Battenberg algorithm as previously described[18]. Briefly, the algorithm phases heterozygous SNPs with use of the 1000 genomes genotypes as a reference panel followed by correcting occasional errors in phasing in regions with low linkage disequilibrium. Segmentation is derived from b-allele frequency (BAF) values. T-tests are performed on the BAFs of each copy number segment to identify whether they correspond to the value resulting from a fully clonal copy number change. If not, the copy number segment is represented as a mixture of 2 different copy number states, with the fraction of cells bearing each copy number state estimated from the average BAF of the heterozygous SNPs in that segment. The segmentation for the

chromosome X in male subjects is processed differently as previously described[26], where copy number segments are called only for the dominant cancer clone. In addition, we applied a non-parametric joint segmentation approach in FACETs[74] to validate the large-scale SCNA callings (Supplementary Method).

**Somatic structural variant calling**. We used the Meerkat algorithm[30] to call somatic SVs and estimate the corresponding genomic positions of breakpoints from recalibrated BAM files. Meerkat has been found to perform better than other previous software in a large analysis across different cancer types[75]. We used parameters adapted to the sequencing depth for both tumor and normal tissue samples and the library insert size. In summary, candidate breakpoints were first found based on soft-clipped and split reads, which requires identifying at least two discordant read pairs, with one read covering the actual breakpoint junction, and then confirmed to be the precise breakpoints by local alignments ('meerkat.pl'). Mutational mechanisms were predicted based on homology and sequencing features ('mechanism.pl'). SVs from tumor genomes were filtered by those in normal genomes. SVs found in simple or satellite repeats were also excluded from the output ('somatic_sv.pl'). The final somatic SVs were annotated as a uniformed format for all breakpoints ('fusions.pl'). We compared the results obtained by Meerkat with those obtained by Novobreak[76] (v1.1.3rc) (Supplementary Method). We opted to retain Meerkat-derived results because they were more conservative and were largely confirmed by laboratory testing. The CCF of SVs in each region was estimated by Svclone[77]; the copy-number subclone information generated by the Battenberg algorithm[18] was used as input for the filter step. To substantially increase the number of variants available for clustering, we applied the coclustering mode to estimate CCF for both SVs and SNVs simultaneously and calculated the average CCF of SVs across regions.

**Validation of somatic structural variants**. We selected four in-frame fusions *MALAT-TFEB, MET-MET* deletion, *STRN-ALK*, and *EWSR1-PATZ1*, for validation by reverse transcription and PCR-based sequencing. The *MALAT-TFEB* and *EWSR1-PATZ1* fusions were validated and confirmed by Sanger sequencing. The other two fusions were not validated because of poor RNA quality from FFPE samples (RIN = 2.6). We selected 381 additional structural variants from pRCC tumors for validation by Ion Torrent PGM Sequencing using a custom AmpliSeq primer pool. We were able to successfully design compatible primers for 303 of them. These included: 87 trunk SVs, 115 internal branch SVs, and 101 terminal branch SVs. 5 SVs failed QC. Among the remaining 298 SVs, 263 (263/298 = 88.3%) were validated at the tumor level and 217 (217/263 = 83%) were validated at clonal level as trunk, internal, or terminal branches. Further details are in the Supplementary Method.

**Somatic mutation calling from deep targeted sequencing data**. We utilized the WGS pipeline to process raw reads, align reads to the reference human genome hg19, and to call somatic SNVs by GATK MuTect2. We then performed multiple mutation filtering and mutation annotation. Given the deep sequencing coverage, we used strict filtering criteria, retaining variants with read depth > = 30 in tumor samples and the number of variant supporting reads ≥ 8. Among the 254 targeted candidate cancer driver genes, we found 67 genes with non-synonymous single nucleotide variant detected by targeted sequencing, 93.6% of which were SNVs called based on WGS data. In contrast, 78.6% of SNVs detected by WGS data were validated by targeted sequencing. High correlation was observed for the variant allele fraction between target sequencing and whole-genome sequencing (Pearson's correlation coefficient = 0.87, P value = 8.54 × 10$^{-88}$).

**Copy-number analysis from genome-wide SNP genotyping data**. Genome Studio (Illumina, Inc.) was used to cluster and normalize raw genotyping data. Both BAF and LogR data were generated and exported for downstream analysis. ASCAT[78] (https://www.crick.ac.uk/peter-van-loo/software/ASCAT) was used to estimate the allele-specific copy numbers without matched normal data. Purity, ploidy, and segmentation data generated by ASCAT were compared to those generated by Battenberg and FACETS (Supplementary Fig. 8).

**Analysis for DNA methylation profiling**. Genome-wide DNA methylation was profiled on Illumina Infinium methylation EPIC arrays (Illumina, San Diego, USA). Methylation of tumor and normal samples was measured according to the manufacturer's instruction at CGR. Raw methylation densities were analyzed using the RnBeads pipeline[79] and the minfi package[80]. In total, we retained 814,408 probes for the downstream analysis. Duplicated samples were selected based on probe intensity, SNP calling rate, and the percentage of failed probes. No batch effects were identified and there were no plating issues. "Functional Normalization"[81], implemented in the minfi R package was used to perform normalization to obtain the final methylation levels (beta value). Hyper- and hypo-methylation were arbitrarily defined by at least 20% in-/decrease relative to the matched normal samples, respectively (Further details in the Supplementary Method).

**Unsupervised clustering of methylation profiles**. We selected the top 1% of probes with the greatest difference between maximum and minimum methylation levels within each tumor. For hierarchical clustering, a Euclidean distance was calculated and Ward's linkage was performed. Normal samples were excluded for the calculation of intratumoral DNA methylation range. Heatmaps were drawn using the superheat (https://github.com/rlbarter/superheat) and ComplexHeatmap R package.

**Measuring intratumoral heterogeneity of SNVs and methylation in genomic regions**. We measured genomic region-specific intratumoral heterogeneity (ITH) of each tumor with at least three samples for DNA methylation levels. DNA methylation variability[82] was calculated as median of the range of probes (maximum methylation level - minimal methylation level) within a genomic region/context among normal samples or within samples in each tumor.

Interindividual variability was analyzed by comparing normal samples from all subjects. The genomic region-specific methylation inter- and intra-tumor heterogeneity was measured by the median methylation variability of involved CpG sitesacross different genomic regions/contexts, including intergenic, 1to5kb, promoters, 5′-UTRs, first exon, exon-intron boundaries, exons, introns, intron-exon boundaries, 3′-UTRs, lncrna_gencode and enhancers_fantom defined in R annotatr package (https://github.com/hhabra/annotatr). The higher the methylation variability, the more ITH observed.

**Statistical analysis**. Statistical analyses were performed using R software (https://www.r-project.org/). Categorical variables were compared using the Fisher's Exact test. Group variables were compared using Wilcoxon rank sum and signed rank test. Comparison of subtypes were by Kruskal–Wallis Test. P values were derived from two-sided tests and those less than 0.05 were considered as statistically significant.

**Reporting Summary**. Further information on research design is available in the Nature Research Reporting Summary linked to this article.

## Data availability

The whole-genome sequencing data, Methylation EPIC data, genotyping data and target-sequencing data have been deposited in the database of Genotypes and Phenotypes (dbGaP) under accession code phs001573.v1.p1; study website.

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

## Acknowledgements

This work was supported by the Intramural Program of the Division of Cancer Epidemiology and Genetics, National Cancer Institute, NIH and utilized the computational resources of the NIH high-performance computational capabilities Biowulf cluster (http://hpc.nih.gov) and DCEG CCAD cluster. We are grateful to the patients and families who contributed to this study and the many investigators who are involved in the NCI-sponsored GEPIKID study of kidney cancer. We also thank the NCI TCGA Program Office for organizational and logistical support, Ms. Preethi Raj for graphical support, and The National Cancer Institute Cancer Genomics Research Laboratory (CGR) for sample preparation and quality control laboratory analyses. L.B.A. is an Abeloff V scholar and he is personally supported by an Alfred P. Sloan Research Fellowship and a Packard Fellowship for Science and Engineering. The research was supported by U.S. Department of Energy National Nuclear Security Administration under Contract No. 89233218CNA000001 and Los Alamos National Laboratory Directed Research and Development Grant, No.20190020DR. DCW is funded by the Li Ka Shing Foundation and the National Institute for Health Research, Oxford Biomedical Research Centre.

## Author Contributions

M.L.P. and M.Constantini conceived the surgical sampling design, collected all samples and organized field activities. B.Z. performed the statistical analysis of phylogenetic trees and supervised the genomic analyses. T.Z. conducted all bioinformatics analyses. J.S., X.H. and D.C.W. helped with the statistical analyses. S.S. reviewed the histological diagnosis of all tumors. M.Constantini and V.P. conducted all clinical examinations and collected clinical data. M.E.D. collected samples and extracted analytes. M.Cardelli analyzed the retrotransposition events. B.S.A., B.O., L.B.A. analyzed mutational signatures and related topography characteristics; K.J., S.B., M.Y., M.W. and B.H. confirmed all laboratory validations. W.Z., J.T. and D.C.W. participated in data interpretation. K.M,B, J.S., and S.C. participated in study conception and data interpretation. S.C. provided resources for the genomics analyses. V.M.F. supervised the field activities and data collection. M.G. performed all surgeries and supervised the sampling collection. M.T.L. conceived the study. B.Z., D.C.W. and M.T.L. discussed the results and implications and wrote the manuscript.

## Competing interests

The authors declare no competing interests.
