## [Peer Review File · Nature Communications]

Reviewers' comments:

Reviewer #1 (Remarks to the Author): Expertise in kidney cancer genomics and clonal evolution

The authors analysed 29 non-clear-cell renal cancers, which are rarer than clear cell tumours (ccRCC) and whose evolutionary dynamics aren't as well described. 25/29 samples were papillary renal cell carcinoma (pRCC). They performed multi-region whole-genome sequencing, targeted deep sequencing of common driver genes, and methylation arrays on a total 124 samples. Analysing the data allowed them to shed light on the evolutionary dynamics and the extent of intra-tumour heterogeneity (ITH) in this disease.

The authors report a lower ITH than in ccRCC on average, with a higher proportion of mutations being shared by all regions. They find that somatic copy number variants (SCNAs) appear to be frequently clonal and thus occurred early during pRCC development, which they confirmed using SNP-array data. This is in agreement with multiple studies in different tumour types that suggest the occurrence of punctuated evolution mediated by copy numbers. This was particularly true for large events, while smaller ones tended to be more often subclonal. The author further made use of the whole-genome sequencing to characterize structural variants (SVs). These were more prominent in one subtype (pRCC2) and tended to be more often subclonal than SCNAs. They also suggest that different callers have different sensitivity according to the mechanism and timing of SV/SCNAs. Mutational signature analyses suggested that most SNVs stemmed from unknown processes, with very few ones explained by aging (signature 1). The authors finally find interesting patterns of congruent evolution, with the epigenetic and genetic divergences between samples being correlated.

This study presents very valuable genetic and epigenetic multi-region data for rarer kidney cancer subtypes, which are analysed thoroughly and elegantly. I do not have any major issue to report regarding this manuscript, with only minor comments listed below.

1. "eight pRCC1 (31%) and three pRCC2 (25%), that had no detected SNVs 141 or indels in previously reported driver genes, even after deep targeted sequencing". I guess the authors mention samples with blank Clonal Mutation Burden (per Mb) in Figure 1c, but some of them display mutations with pretty high CCFs. From Figure 1c, although I do find 5 samples from 3 pRCC2 patients with no SNV/indel (pRCC2_1799_02_T01&T02, pRCC2_1410_02_T01&T02 and pRCC2_1396_01_T01), I only find 4 pRCC1 samples with no SNV or indel (RCC1_1699_01_T01, pRCC1_1654_01_T01, RCC1_1501_01_T01 and pRCC1_1546_01_T01). If the unique SNV/indels found in the other four single-sampled tumours are not considered clonal, this should be more precisely annotated.
2. Perhaps it would be good to have a quick introductory sentence to mention how pRCC subtypes 1 and 2 are defined, as despite the mentioned prognosis difference it is not obvious where the differences lie.
3. Are the differences in SNV/indel rates significant between cancer subtypes? For instance, if they are individualized per patient and compared using a non-parametric test.
4. Are the number of clones detected by DPclust dependent on the number of samples sequenced per patient? Can it be transformed into a per-patient normalised metric to compare ITH levels between subtypes?
5. The authors report variegated subclones in distant regions, with cell motility as the likely explanation. I disagree with this being the only explanation. It may be due to ancient but slow-proliferating clones getting separated as others expand, such as was illustrated by the "big bang" paper (Sottoriva et al., Nature Genetics, 2015). This should also be mentioned as one of the possible explanations.
6. Figure S8 is not really showing what it's referenced for in the text (arm-level SCNA clonality). Do the authors mean S7? Figure S8 is interesting and show the agreement between FACETS

(WGS) and ASCAT (SNP array), which probably should be mentioned even if briefly in the text.

7. In Figure S7, cdRCC_1929_03_T02 seems to display copy-neutral LOH on entire genome. I find it hard to believe and could be due to ploidy/purity miscalculations prior to SCNA calling. It however does not impact any of the authors' reported findings.

8. I found the SV/SCNA comparison paragraph (the one mentioning Fig S17) and the methodology a bit obscure. No p-values are given either. I'd suggest rewriting or removing (it doesn't contribute much to the knowledge of the disease, which is the article's main aim).

9. I find it puzzling that the aging mutational signature's contribution (sig 1) is so low (3.5% of the total), given the high proportion of clonal SNVs. Most, or at least a fair share of the clonal mutations should be due to aging. Are signatures 5 and 40 patterns similar to signature 1? I understand that the authors did not aim to solve where the unknown signatures 5 and 40 stem from, but it would be interesting to see this elaborated on in the discussion.

10. How exactly was "variability" quantified in methylation-based ITH analyses? Standard deviation? Coefficient of variation? This should be detailed further.

11. Pettiest comment: The figures are stylish and nice-looking overall, but I found the purple/yellow combination in Figure 3b very brutal!

Pierre Martinez.

Reviewer #2 (Remarks to the Author): Expertise in kidney cancer epigenetics

This study was designed to answer an important question concerning intra-tumoral heterogeneity in kidney tumors that are not clear cell renal cell carcinomas (ccRCCs). A series of non-clear cell RCCs with an emphasis on papillary renal cell carcinoma (pRCC) were multi-regionally sampled and evaluated for DNA mutation, chromosomal alterations, and DNA methylation. The authors identified a distinct difference between clear cell RCC that show considerable ITH and these non-clear cell RCC subtypes that largely demonstrated clonal evolution with little ITH. These observations were slightly impeded by the number of samples available for analysis and fact that only a single example was available for several RCC subtypes. Potentially, this data would be very important as it could indicate whether multiple biopsies are necessary for accurate characterization of non-ccRCC tumors and the likelihood of resistant clones arising from a heterogeneous primary tumor in response to therapy. Currently, the study has some limitations and my comments are listed below:

Major Comments:

1. The epigenomic analysis within this study was limited to simply DNA methylation. Further analyses such as MNase-seq or ATAC-seq that evaluate chromatin accessibility could be used to provide a more thorough investigation of the epigenomic state of these tumors. The title of the study should be altered to reflect that DNA methylation was analyzed rather than a complete epigenomic analysis.

2. The methods and figure 1b state that at least 23 pRCC1 tumors received targeted sequencing and SNP genotyping, but most of the study concentrates on the 13 pRCC1 tumors that also received whole genome sequencing and methylation analysis. Could the authors highlight the importance of the data from the additional 10 pRCC1 tumors?

3. Three of the pRCC2 did not receive any methylation analysis. The authors need to state which were not analyzed and indicate on figure 1c that the methylation levels were not assessed in 3 tumors. Where these tumors not analyzed for a specific reason?

4. Four of the 13 analyzed pRCC1 tumors (pRCC1-1689, pRCC1-1416, pRCC1-1699, and pRCC1-1472) lack the classic chromosomal changes associated with pRCC1 such as gain of chromosomes 7, and 17. Did these tumors demonstrate unusual histologies or unusual IHC staining patterns? Similarly, three of the 12 pRCC2 tumors (pRCC2-1824, pRCC2-1568, pRCC2-1494) demonstrate the classic gain of chromosomes 7 and 17 associated with pRCC1, these tumors were definitely

pRCC2 tumors and not high-grade, more differentiated pRCC1 tumors?

5. Sample pRCC2-1410 has the classic MALAT1-TFEB gene fusion and so should be classified as a single example of a MiT family translocation renal cell carcinoma (TRCC). TRCCs can present with a histology that is described as type 2 papillary but are now considered a subtype of their own. Some comment on this needs to be added to the study as they would not necessarily be expected to show the same patterns of genomic change as pRCC2 tumors.

6. The evaluation of other types of rare kidney cancer is commendable but problematic as in each case only a single tumor was evaluated. It is hard to gauge the value of the data from a single sample, if a single clear cell RCC was taken at random there is a ~30% chance that it would show no ITH, but this would not be a great reflection of ccRCCs in general. It is interesting that none of the others showed ITH at the DNA mutation level, but the rSRC sample did show ITH for chromosomal level changes (Figure 3) yet it is hard to conclude much from this. Combining the "other RCCs" is inadvisable as they are completely different types of tumor and should not be done. The actual mutations and chromosomal changes in those samples are interesting and could be more prominently discussed in the study, rather than their degree of ITH. Additional of any further samples with the same histologies would be very beneficial.

7. An impressive number of pRCC2 tumors had multi-regional sequencing, but the number of pRCC1 tumors with 3 or more sampled regions is very small (n=3) and only represents low stage tumors (stage 1 or 2). Similarly, the metastatic analysis is limited to a single pRCC2 and a single rSRC tumor, which limits the relevance of the observations. Addition of more samples would be extremely beneficial, especially of higher stage pRCC1 tumors and any other metastases.

8. Eight of the pRCC1 tumors, 2 of the pRCC2 tumors and a mix-RCC tumor only had single samplings performed. Could the authors highlight how the input these samples have in the study?

9. Although the levels of ITH were milder than in ccRCC, did ITH correlate with the size of tumor in either pRCC2 or pRCC1? pRCC tumors can have varying histologic features within them, did the histologic analysis of each samples segment within any tumor show distinct differences and was this reflected in any degree of ITH?

Minor Comments:

In figure 1, two labels reading "Papillary type 2" and "Other" seem to have been additionally placed between sections b and c that need to be removed.

In figure 1c, the colors used to designate clonal or subclonal changes are too similar and more distinct colors should be used.

In line 140 it says "... eight pRCC1 (31%).." I think the percentage must be wrong here as I could not repeat it?

Can the authors list the MET mutations that were found in the main text and state whether they occurred in the tyrosine kinase domain of MET?

The authors state there are reports "that ~90% of pRCC do not harbor detectable pathogenic changes in any driver genes". This seems very high, was that true within the TCGA analysis of pRCC that performed comprehensive analysis of genomic changes?

Reviewer #3 (Remarks to the Author): Expertise in phylogenetic analysis

This is a well executed study of a relatively understudied group of cancers, with findings that are scientifically interesting and seemingly have translational potential. The work is thorough, with good consideration for appropriate controls and with the main findings well supported by multiple lines of evidence. The findings are likely to be of interest to the cancer research community generally, those specifically working on renal cancers, and to the community studying clonal evolution in cancers in particular. The study leads to quite a few intriguing observations about these cancers, notably the overwhelmingly truncal placement of large somatic copy number alterations (SCNAs) and single nucleotide variants (SNVs) in contrast to largely subclonal small structural variants (SVs) and focal SCNAs, a preponderance of SNVs from a small number of

mutation signatures, high clonal mixing and likely mobility, and substantial differences in variant rates across tumor subtypes and by tumor within the subtypes. Beyond the specific system, the work contributed nicely to a growing picture of the heterogeneity of mutation mechanisms and other features of clonal evolution across cancer tissues of origin and subtypes.

The study is nicely designed, with several features that make it notable relative to most tumor genomic studies to date. The authors examine a wide variety of variation types derivable from whole genome sequence (WGS) sequencing, as well as several complementary analyses such as methylation. The multiregion sequencing strategy is well suited for a clonal evolution study, even though it is still relatively rare in large tumor genomics studies to date. The computational analysis, like the experimental, is thorough and nicely executed. The paper does a great job of exploring many lines of evidence one can extract from the raw genomic data, with excellent consideration for validation of the key conclusions of the sequencing work. The software analysis pipeline is appropriate for the problem and although alternatives might have been considered for some steps, the choices are all defensible. In particular, the phylogeny method chosen is appropriate for the data and study design. The fine details of inferred phylogenies can be expected to have some uncertainty, particularly in precise definitions of subclones and branch ordering near the leaves, but differences on these points would not change the major conclusions of the paper. I do not see any likely grounds to question the major conclusions, which are well supported by the main sequencing study and various additional validations.

Beyond the main points in the above summary, just a couple of issues raise significant questions for me:

1. Probably the main point on which the study might be criticized is that it is essentially descriptive rather than mechanistic. There are some intriguing hints about mechanism, e.g., in the mention of frequent germline SNPs in known driver genes, but no clear explanations for the main findings. For example, the work does not compellingly establish why variant types segregate between truncal large SNCAs and SNVs vs. SVs that are largely found towards the leaves. There is a generic hypothesis based on conclusions of prior work, but there is no obvious way to test it. Likewise, there are some intriguing results from the mutational signature analysis, but the major SNV drivers are signatures of unknown origin and the observation that signatures are quite conserved between primary and metastasis but quite variable by tumor has no explanation. I do not think this is a fatal problem for the paper, as the descriptive results are quite interesting in their own right and it is understandable that description has to come first before some of the mysteries can be resolved in later work. Still, the issue might bear some further examination, at least to rule out any obvious connections between the observed patterns of accumulating mutations by tumor type and the major recurrent somatic or germline driver mutations observed in the different tumor types.

2. Although the multiregion study design is I think generally quite well done, the one notable point on which I would have wished to have seen more would be a comparison between tumor and adjacent normal tissue, using distant normal or blood normal as a reference. If I understood correctly, the study used adjacent normal tissue as the reference for all tumor samples, and thus would not be able to discriminate between mutation processes active in the tumor vs. the adjacent normal tissue. It would seem that such a comparison could shed light on a number of unknowns in the current analysis. For example, which of the mutation signatures in the tumors are recapitulated in adjacent normal tissue and which are tumor-specific? Do these normal tissue signatures differ by tumor type the subject developed? Are there changes in signatures during tumor development, which could distort timing inferences of events in the long trunks in the phylogenies? The latter could make the difference between interpreting the data as reflective of a sudden catastrophic recent change in SCNAs following a long mutation process typical of renal tissue in these patients or of an early catastrophe followed by a period of dormancy with elevated mutation rate.

I also noted a few minor errors:

1. Line 439: "SNVsand"
2. Line 663: "Merkaat"
3. Line 670: Missing close quote on ""mechanism.pl"
4. Line 683: "selectedfour"
5. Line 744: Missing space in "levels.DNA"
6. Line 758: Missing space in "Test.P"
7. References: There seem to be consistently missing spaces between the journal title and volume.
8. Supplementary Material, lines 109-110: There seems to be some oddness with the font size here

Reviewer #1 (Remarks to the Author): Expertise in kidney cancer genomics and clonal evolution

The authors analysed 29 non-clear-cell renal cancers, which are rarer than clear cell tumours (ccRCC) and whose evolutionary dynamics aren't as well described. 25/29 samples were papillary renal cell carcinoma (pRCC). They performed multi-region whole-genome sequencing, targeted deep sequencing of common driver genes, and methylation arrays on a total 124 samples. Analysing the data allowed them to shed light on the evolutionary dynamics and the extent of intra-tumour heterogeneity (ITH) in this disease.

The authors report a lower ITH than in ccRCC on average, with a higher proportion of mutations being shared by all regions. They find that somatic copy number variants (SCNAs) appear to be frequently clonal and thus occurred early during pRCC development, which they confirmed using SNP-array data. This is in agreement with multiple studies in different tumour types that suggest the occurrence of punctuated evolution mediated by copy numbers. This was particularly true for large events, while smaller ones tended to be more often subclonal. The author further made use of the whole-genome sequencing to characterize structural variants (SVs). These were more prominent in one subtype (pRCC2) and tended to be more often subclonal than SCNAs. They also suggest that different callers have different sensitivity according to the mechanism and timing of SV/SCNAs. Mutational signature analyses suggested that most SNVs stemmed from unknown processes, with very few ones explained by aging

(signature 1). The authors finally find interesting patterns of congruent evolution, with the epigenetic and genetic divergences between samples being correlated.

This study presents very valuable genetic and epigenetic multi-region data for rarer kidney cancer subtypes, which are analysed thoroughly and elegantly. I do not have any major issue to report regarding this manuscript, with only minor comments listed below.

1. "eight pRCC1 (31%) and three pRCC2 (25%), that had no detected SNVs 141 or indels in previously reported driver genes, even after deep targeted sequencing". I guess the authors mention samples with blank Clonal Mutation Burden (per Mb) in Figure 1c, but some of them display mutations with pretty high CCFs. From Figure 1c, although I do find 5 samples from 3 pRCC2 patients with no SNV/indel (pRCC2_1799_02_T01&T02, pRCC2_1410_02_T01&T02 and pRCC2_1396_01_T01), I only find 4 pRCC1 samples with no SNV or indel (RCC1_1699_01_T01, pRCC1_1654_01_T01, RCC1_1501_01_T01 and pRCC1_1546_01_T01). If the unique SNV/indels found in the other four single-sampled tumours are not considered clonal, this should be more precisely annotated.

Response: We thank the reviewer for this comment. It was an oversight. There were four, not eight, pRCC1 tumors (4/13=31%) with no detected SNVs or indel. As correctly stated by the reviewer, the four pRCC1 tumors include RCC1_1699_01, pRCC1_1654_01, RCC1_1501_01 and pRCC1_1546_01 and the three pRCC2 tumors include pRCC2_1799_02, pRCC2_1410 and pRCC2_1396_01. We have revised the manuscript accordingly.

The Clonal Mutation Burden (per Mb) cells is blank for eight pRCC1 and three pRCC2 because these subjects have one sampled region only and we require more than one sampled region to classify

clonal/subclonal mutations. To avoid confusion, we annotated these cells as “N/A” instead than leaving them blank, and added a new row to indicate no SNVs or indel in cancer driver genes.

2. Perhaps it would be good to have a quick introductory sentence to mention how pRCC subtypes 1 and 2 are defined, as despite the mentioned prognosis difference it is not obvious where the differences lie.

We added the morphological description of the two types as follows: “Papillary RCC are traditionally distinct into 2 types: a) Type 1 with papillae covered by smaller cells with scant amphophilic cytoplasm and single cell layer, and b) Type 2 with large tumor cells, often with high nuclear grade, eosinophilic cytoplasm and nuclear pseudostratification (Delahunt et al., Mod Pathol 1997; Jiang et al., Am J Pathol 1998; Diagnostic pathology, 2016)

3. Are the differences in SNV/indel rates significant *between* cancer subtypes? For instance, if they are individualized per patient and compared using a non-parametric test.

Response: SNV rate is significantly higher in pRCC2 compared to pRCC1 (Wilcoxon test $P=0.03$), but not the indel rate ($P=0.65$). No comparison can be made for the other subtypes because of the small numbers. We added the p-values of non-parametric Wilcoxon test in the Results section (page 5) as suggested and Discussion (last paragraph).

4. Are the number of clones detected by DPCLust dependent on the number of samples sequenced per patient? Can it be transformed into a per-patient normalised metric to compare ITH levels between subtypes?

Response: We thank the reviewer for this comment. Since ITH can be influenced by the number of samples sequenced per tumor, we used a recently proposed ITH metric, average pairwise ITH or APITH (Hua et al., 2019, medRxiv), to compare pRCC1 and pRCC2 ITH. APITH is defined as the average genomic distance across all pairs of samples per tumor and does not depend on the overall number of samples per tumor. We found that APITH of pRCC2 (mean=26.66) is higher than APITH of pRCC1 (mean=16.20, $P=0.03$).

We added these new results to the main text in both the Results (page 6) and Discussion (page 12) sections.

5. The authors report variegated subclones in distant regions, with cell motility as the likely explanation. I disagree with this being the only explanation. It may be due to ancient but slow-proliferating clones getting separated as others expand, such as was illustrated by the “big bang” paper (Sottoriva et al., Nature Genetics, 2015). This should also be mentioned as one of the possible explanations.

Response: We agree with the reviewer that our observation may be due to a quick expansion as described in the “big bang” paper. We added this comment in the Result section (page 7).

6. Figure S8 is not really showing what it's referenced for in the text (arm-level SCNA clonality). Do the authors mean S7? Figure S8 is interesting and show the agreement between FACETS (WGS) and ASCAT (SNP array), which probably should be mentioned even if briefly in the text.

Response: The reviewer is correct. The previous Figure S8 shows the agreement between FACETS (WGS) and ASCAT (SNP array) in terms of ploidy and purity. Following the reviewer's suggestion, we have now added a new panel to Figure S8 comparing arm level SNCA events across platforms, which shows that >95% SCNA segments were identified by both FACETS (WGS) and ASCAT (SNP array).

7. In Figure S7, cdRCC_1929_03_T02 seems to display copy-neutral LOH on entire genome. I find it hard to believe and could be due to ploidy/purity miscalculations prior to SCNA calling. It however does not impact any of the authors' reported findings.

Response: We checked the copy number of sample cdRCC_1929_03_T02 and found that FACETS over-estimated the ploidy of this sample, which is two times higher than that by ASCAT or Battenberg. We correct it in Figure S7.

8. I found the SV/SCNA comparison paragraph (the one mentioning Fig S17) and the methodology a bit obscure. No pvalues are given either. I'd suggest rewriting or removing (it doesn't contribute much to the knowledge of the disease, which is the article's main aim).

Response: we appreciate the feedback and moved the SV/SCNA comparison to the supplementary materials.

9. I find it puzzling that the aging mutational signature's contribution (sig 1) is so low (3.5% of the total), given the high proportion of clonal SNVs. Most, or at least a fair share of the clonal mutations should be due to aging. Are signatures 5 and 40 patterns similar to signature 1? I understand that the authors did not aim to solve where the unknown signatures 5 and 40 stem from, but it would be interesting to see this elaborated on in the discussion.

Response: This is a good point. In our study, most of the mutations are contributed by Single Base Substitution (SBS) signatures 5 and 40. The signature 40 is a relative "flat" signature like signature 5 (cosine similarity: 0.83); in contrast, the signature 1 shows peaks for C>T at the CpG motif. Hence, signatures 5 and 40 patterns are not similar to signature 1.

We found that the number of clonal mutations assigned to signature 1, 5 or 40 is significantly associated with age at diagnosis (SBS1 vs age: Pearson's correlation coefficient (R)= 0.46, P= 0.013; SBS5 vs age: R= 0.40; P= 0.033; SBS40 vs age: R=0.48, P= 0.009), while the number of subclonal mutations assigned to signature 1, 5 or 40 is not. See the plots below. We added these findings to the Results (page 10 and Supplementary Figure S19a and S19b).

Scatter plots of number of clonal mutations assigned to signatures 1, 5, and 40 with age at diagnosis.

Scatter plots of number of subclonal mutations assigned to signatures 1, 5, and 40 with age at diagnosis.

10. How exactly was “variability” quantified in methylation-based ITH analyses? Standard deviation? Coefficient of variation? This should be detailed further.

Response: Following Brocks et al. (Cell Rep. 2014), the DNA methylation variability for ITH analyses was quantified based on the range of methylation probes for multiple samples within a tumor. First, the range of each probe (maximum methylation level - minimal methylation level) was obtained; and then methylation-based variability was calculated as the median of range across all probes within a genomic region, e.g. promoter region. We added more details to the Methods section (pages 22 and 23).

11. Pettiest comment: The figures are stylish and nice-looking overall, but I found the purple/yellow combination in Figure 3b very brutal!

Response: we have changed the color of Figure 3b to be consistent with the other figures.

Pierre Martinez.

Reviewer #2 (Remarks to the Author): Expertise in kidney cancer epigenetics

This study was designed to answer an important question concerning intra-tumoral heterogeneity in kidney tumors that are not clear cell renal cell carcinomas (ccRCCs). A series of non-clear cell RCCs with an emphasis on papillary renal cell carcinoma (pRCC) were multi-regionally sampled and evaluated for DNA mutation, chromosomal alterations, and DNA methylation. The authors identified a distinct difference between clear cell RCC that show considerable ITH and these non-clear cell RCC subtypes that largely demonstrated clonal evolution with little ITH. These observations were slightly impeded by the number of samples available for analysis and fact that only a single example was available for several RCC subtypes. Potentially, this data would be very important as it could indicate whether multiple biopsies are necessary for accurate characterization of non-ccRCC tumors and the likelihood of resistant clones arising from a heterogeneous primary tumor in response to therapy. Currently, the study has some limitations and my comments are listed below:

Major Comments:

1. The epigenomic analysis within this study was limited to simply DNA methylation. Further analyses such as MNase-seq or ATAC-seq that evaluate chromatin accessibility could be used to provide a more thorough investigation of the epigenomic state of these tumors. The title of the study should be altered to reflect that DNA methylation was analyzed rather than a complete epigenomic analysis.

Response: We agree with the reviewer that further analysis of chromatin accessibility would add more insight into the epigenomic evolution. We had already mentioned in the Abstract that we had focused our epigenetic analysis to DNA methylation. We have now emphasized this point in the Introduction (page 4) and Results (page 11).

2. The methods and figure 1b state that at least 23 pRCC1 tumors received targeted sequencing and SNP genotyping, but most of the study concentrates on the 13 pRCC1 tumors that also received whole genome sequencing and methylation analysis. Could the authors highlight the importance of the data from the additional 10 pRCC1 tumors?

Response: Although we had only a single sample from 10 pRCC1 tumors, we conducted targeted sequencing on these tumors to improve our knowledge of cancer driver genes in this rare cancer type. Among these 10 tumors, we found 2 ATM, 2 MET, 1 IDH1, 1 EP300, 1 KMT2A, 1 KMA2C, and 1 NFE2L2 mutations. We have specified these findings in the Results section (page 5).

3. Three of the pRCC2 did not receive any methylation analysis. The authors need to state which were not analyzed and indicate on figure 1c that the methylation levels were not assessed in 3 tumors. Where these tumors not analyzed for a specific reason?

Response: These three samples had not enough DNA to conduct methylation array analysis. We added one row in figure 1c indicating presence/absence of DNA methylation data.

4. Four of the 13 analyzed pRCC1 tumors (pRCC1-1689, pRCC1-1416, pRCC1-1699, and pRCC1-1472) lack the classic chromosomal changes associated with pRCC1 such as gain of chromosomes 7, and 17. Did these tumors demonstrate unusual histologies or unusual IHC staining patterns? Similarly, three of the 12 pRCC2 tumors (pRCC2-1824, pRCC2-1568, pRCC2-1494) demonstrate the classic gain of chromosomes 7 and 17 associated with pRCC1, these tumors were definitely pRCC2 tumors and not high-grade, more differentiated pRCC1 tumors?

Response: The definition of pRCC1 and pRCC2 was based on morphological evaluation. We relied on centralized pathology review for all these tumors by an expert uropathologist, who confirmed the classification of tumor subtypes. We also reviewed again the SCNAs identified in our samples, and confirmed the results previously reported in the original manuscript. However, we cannot exclude the possibility that these tumors have mixed histologies (e.g., papillary types 1 and 2) in sections that were not available for histological review. We have added this point in the Online Methods (page 17).

5. Sample pRCC2-1410 has the classic MALAT1-TFEB gene fusion and so should be classified as a single example of a MiT family translocation renal cell carcinoma (TRCC). TRCCs can present with a histology that is described as type 2 papillary but are now considered a subtype of their own. Some comment on this needs to be added to the study as they would not necessarily be expected to show the same patterns of genomic change as pRCC2 tumors.

Response: We thank the reviewer for this important comment. We agree that this subtype could be considered separately from pRCC2, although the morphological characteristics are the same. We added a comment on this in the Results section (pages 8 and 9). "We note that one tumor (pRCC2-1410), which had the morphological features of pRCC2, showed the classic MALAT1-TFEB gene fusion. Thus, it should be considered a MiT family translocation renal cell carcinoma (TRCC) (Kauffman et al, Nat Rev Urol, 2014; Calio et al. Cancers, 2019). As expected for this subtype, this patient had a good prognosis (long survival and no metastasis)"

6. The evaluation of other types of rare kidney cancer is commendable but problematic as in each case only a single tumor was evaluated. It is hard to gauge the value of the data from a single sample, if a single clear cell RCC was taken at random there is a ~30% chance that it would show no ITH, but this would not be a great reflection of ccRCCs in general. It is interesting that none of the others showed ITH at the DNA mutation level, but the rSRC sample did show ITH for chromosomal level changes (Figure 3) yet it is hard to conclude much from this. Combining the "other RCCs" is inadvisable as they are completely different types of tumor and should not be done. The actual mutations and chromosomal changes in those samples are interesting and could be more prominently discussed in the study, rather than their degree of ITH. Additional of any further samples with the same histologies would be very beneficial.

Response: We agree with the reviewer that to increase the sample size would be beneficial. Unfortunately, given the rarity of these tumors, we are unable to increase their sample size. We had

reported the molecular features (SCNAs, fusions, SNVs in cancer driver genes) for each of these rare kidney cancer subtypes in the original manuscript and described them separately in every figure. We have now reported the mutational burden for each rare cancer subtype separately, as suggested by the reviewer (page 5).

7. An impressive number of pRCC2 tumors had multi-regional sequencing, but the number of pRCC1 tumors with 3 or more sampled regions is very small (n=3) and only represents low stage tumors (stage 1 or 2). Similarly, the metastatic analysis is limited to a single pRCC2 and a single rSRC tumor, which limits the relevance of the observations. Addition of more samples would be extremely beneficial, especially of higher stage pRCC1 tumors and any other metastases.

Response: Unfortunately, we have no sequencing data from additional pRCC1 multi-region samples or metastatic samples from any subtype to be added to our study. Given the good prognosis, pRCC1 tumors had no metastases and tended to be smaller in size and with early stage; thus only one or two samples could be collected from several pRCC1 tumors.

8. Eight of the pRCC1 tumors, 2 of the pRCC2 tumors and a mix-RCC tumor only had single samplings performed. Could the authors highlight how the input these samples have in the study?

Response: Although we had no more than one single sample from these tumors, we thought that sequencing them could contribute to a better knowledge of the tumor landscape of these tumor types. Thus, we reported mutation rate, frequency of mutations in cancer driver genes and any other feature from these tumors.

9. Although the levels of ITH were milder than in ccRCC, did ITH correlate with the size of tumor in either pRCC2 or pRCC1? pRCC tumors can have varying histologic features within them, did the histologic analysis of each samples segment within any tumor show distinct differences and was this reflected in any degree of ITH?

Response:

Following the reviewer's suggestions, we analyzed the correlation between tumor size and ITH. We found no significant association between the size of tumor and ITH (measured by APITH which is independent from the number of samples per tumor, $P=0.38$, all tumors; $P=0.81$, pRCC1; $P=0.46$, pRCC2). We added this result in the manuscript (page 6).

Although we used H&E-stained formalin-fixed samples for the overall tumor diagnosis, we used frozen samples for whole genome sequencing (WGS). We histologically reviewed each frozen samples that was used for WGS to estimate the percentage of tumor cells. However, subtle morphological differences cannot be assessed well using frozen sections. Thus, we cannot perform an analysis linking WGS data with subtle histological features across regions from the same tumors.

Minor Comments:

In figure 1, two labels reading “Papillary type 2” and “Other” seem to have been additionally placed between sections b and c that need to be removed.

Response: We thanks the reviewer for pointing out to this mistake. We have corrected it in the revised version.

In figure 1c, the colors used to designate clonal or subclonal changes are too similar and more distinct colors should be used.

Response: More distinct colors have been used.

In line 140 it says “.., eight pRCC1 (31%)..” I think the percentage must be wrong here as I could not repeat it?: percentage based on total number of subjects not samples

Response: There should be four not eight tumors. We corrected this in the main text (Results, page 5).

Can the authors list the MET mutations that were found in the main text and state whether they occurred in the tyrosine kinase domain of MET?

Response: All mutations occurred in the tyrosine kinase domain (see figure below). We added this information in the Results section, page 5.

The authors state there are reports “that ~90% of pRCC do no harbor detectable pathogenic changes in any driver genes”. This seems very high, was that true within the TCGA analysis of pRCC that performed comprehensive analysis of genomic changes?

Response: We thank the reviewer for pointing out this mistake. The number should be $7/31=22.6\%$ in the study by Kovac et al. (Nat. Commun. 2015). We also checked the TCGA analysis of pRCC (NEJM,2016), overall ~23% of pRCC had no driver events. We added this information in the Results section (page 5).

Reviewer #3 (Remarks to the Author): Expertise in phylogenetic analysis

This is a well executed study of a relatively understudied group of cancers, with findings that are scientifically interesting and seemingly have translational potential. The work is thorough, with good consideration for appropriate controls and with the main findings well supported by multiple lines of evidence. The findings are likely to be of interest to the cancer research community generally, those specifically working on renal cancers, and to the community studying clonal evolution in cancers in particular. The study leads to quite a few intriguing observations about these cancers, notably the overwhelmingly truncal placement of large somatic copy number alterations (SCNAs) and single nucleotide variants (SNVs) in contrast to largely subclonal small structural variants (SVs) and focal SCNAs, a preponderance of SNVs from a small number of mutation signatures, high clonal mixing and likely mobility, and substantial differences in variant rates across tumor subtypes and by tumor within the subtypes. Beyond the specific system, the work contributed nicely to a growing picture of the heterogeneity of mutation mechanisms and other features of clonal evolution across cancer tissues of origin and subtypes.

The study is nicely designed, with several features that make it notable relative to most tumor genomic studies to date. The authors examine a wide variety of variation types derivable from whole genome sequence (WGS) sequencing, as well as several complementary analyses such as methylation. The multiregion sequencing strategy is well suited for a clonal evolution study, even though it is still relatively rare in large tumor genomics studies to date. The computational analysis, like the experimental, is thorough and nicely executed. The paper does a great job of exploring many lines of evidence one can extract from the raw genomic data, with excellent consideration for validation of the key conclusions of the sequencing work. The software analysis pipeline is appropriate for the problem and although alternatives might have been considered for some steps, the choices are all defensible. In particular, the phylogeny method chosen is appropriate for the data and study design. The fine details of inferred phylogenies can be expected to have some uncertainty, particularly in precise definitions of subclones and branch ordering near the leaves, but differences on these points would not change the major conclusions of the paper. I do not see any likely grounds to question the major conclusions, which are well supported by the main sequencing study and various additional validations.

Beyond the main points in the above summary, just a couple of issues raise significant questions for me:

1. Probably the main point on which the study might be criticized is that it is essentially descriptive rather than mechanistic. There are some intriguing hints about mechanism, e.g., in the mention of frequent germline SNPs in known driver genes, but no clear explanations for the main findings. For example, the work does not compellingly establish why variant types segregate between truncal large SCNAs and SNVs vs. SVs that are largely found towards the leaves. There is a generic hypothesis based

on conclusions of prior work, but there is no obvious way to test it. Likewise, there are some intriguing results from the mutational signature analysis, but the major SNV drivers are signatures of unknown origin and the observation that signatures are quite conserved between primary and metastasis but quite variable by tumor has no explanation. I do not think this is a fatal problem for the paper, as the descriptive results are quite interesting in their own right and it is understandable that description has to come first before some of the mysteries can be resolved in later work. Still, the issue might bear some further examination, at least to rule out any obvious connections between the observed patterns of accumulating mutations by tumor type and the major recurrent somatic or germline driver mutations observed in the different tumor types.

Response: As reviewer pointed out, the dominant signatures in our study were COSMIC signatures SBS5 and SBS40, of which the etiology is unknown. Signatures 5 has been previously reported as clock-like signature; as expected we found that signature 40 is very similar to signature 5 (cosine similarity: 0.83). Clonal mutations in signatures 1, 5 and 40 were significantly associated with the age of diagnosis. See more details in response 9 to the first reviewer. We added these results in the Results section (page 10 and Supplementary Figures S19a and S19b) and the following paragraph in the Discussion.

Signatures SBS5 and SBS40 accounted for 92.5% of all somatic mutations observed in pRCC. High frequency of signature SBS40 has been found in kidney cancer in previous studies, possibly due to the organ's cells constant contact with mutagens during the blood filtration process (Alexandrov, L. et al. bioRxiv, 2018). Both signatures have unknown etiology, but signature SBS5 has been associated with age at diagnosis across most human cancers (Alexandrov, L.B. et al, Nat Genet, 2015). These "flat" signatures are correlated to each other and likely harbor common mutation components³¹. In our study, clonal mutations attributed to signatures 1, 5, and 40 were all significantly correlated with age of diagnosis, suggesting that they may be the result of a lifetime accumulation of mutations. Future experimental studies are necessary to investigate the molecular and mutational underpinning of signatures 5 and 40. Notably, among the 29 subjects with WGS data, 13 were never smokers, 4 current smokers, 6 former smokers and 6 had unknown smoking status. However, we found no tobacco smoking signature SBS4, as previously observed in kidney and bladder cancers (Alexandrov, L.B. et al. Science, 2016).

We also added some discussion on the mechanisms of SV formation on pages 11 and 12 of the Discussion as follows:

The mechanisms of SVs formation are largely unknown. A landscape description of breast cancer (Nik-Zainal, S. et al. Nature, 2016) and a recent structural variant analysis in PCAWG (Gundem, G. et al. Nature, 2015) identified different signatures of structural variants, separated by size. Taken all together, these findings suggest that there are different mutational and repair processes operating at different scales and future research should be directed towards further elucidating the causal mechanisms.

2. Although the multiregion study design is I think generally quite well done, the one notable point on which I would have wished to have seen more would be a comparison between tumor and adjacent

normal tissue, using distant normal or blood normal as a reference. If I understood correctly, the study used adjacent normal tissue as the reference for all tumor samples, and thus would not be able to discriminate between mutation processes active in the tumor vs. the adjacent normal tissue. It would seem that such a comparison could shed light on a number of unknowns in the current analysis. For example, which of the mutation signatures in the tumors are recapitulated in adjacent normal tissue and which are tumor-specific? Do these normal tissue signatures differ by tumor type the subject developed? Are there changes in signatures during tumor development, which could distort timing inferences of events in the long trunks in the phylogenies? The latter could make the difference between interpreting the data as reflective of a sudden catastrophic recent change in SCNAs following a long mutation process typical of renal tissue in these patients or of an early catastrophe followed by a period of dormancy with elevated mutation rate.

Response: We agree with the reviewer that distinguishing somatic processes in normal and cancerous tissues is of great interest. We did not have blood samples available in this study, but intend to explore this in future studies, by sequencing multiple normal samples and blood samples from individual patients. As we did not estimate mutation rates in normal tissue in this study, we are not able to study the progression of mutational signatures from adjacent normal tissue to the tumor tissue or link directly tree branch lengths to chronological time. However, we were able to distinguish clonal and subclonal events. From this, it is apparent that driver SNVs and large SCNAs occur earlier than small SCNAs. In addition we were able to order the occurrence of driver mutations relative to somatic copy number gains or loss of heterozygosity (LOH; Nik-Zainal, S. et al., Cell, 2012; Shinde, J. et al., Bioinformatics, 2018) and were able to infer the timing of some driver mutations (Table S13). For example, the SMARCB1 p.R373T mutation occurred earlier than the 22q LOH in pRCC2_1824_13_T08, and the truncated mutation KMT2C p.S789 occurred later than the chr7 amplification in pRCC2_1494. We added these findings in the Results section on page 8.*

I also noted a few minor errors:

1. Line 439: "SNVsand"
2. Line 663: "Merkaat"
3. Line 670: Missing close quote on ""mechanism.pl"
4. Line 683: "selectedfour"
5. Line 744: Missing space in "levels.DNA"
6. Line 758: Missing space in "Test.P"
7. References: There seem to be consistently missing spaces between the journal title and volume.

8. Supplementary Material, lines 109-110: There seems to be some oddness with the font size here

Response: We thank the reviewer for these comments. These errors have been corrected. The “consistently missing spaces between the journal title and volume” is due to convert format from WORD to PDF. We will upload a PDF format instead.

Reviewers' comments:

Reviewer #1 (Remarks to the Author):

The authors have satisfactorily addressed all my comments. I now recommend the article for publication.

Pierre Martinez.

Reviewer #2 (Remarks to the Author):

The authors have altered some of the text to accommodate my comments and this is appreciated. It is understandable that the authors do not necessarily have any additional samples to add to their study or any more samplings from the existing tumors to perform any further analysis, but this does still substantially limit the conclusions that can be drawn from the study. There is still no analysis of the higher stage more aggressive pRCC1 tumors that are the ones that might show ITH and will metastasize and are probably the most worth analyzing, but this is a limitation of the materials available. The single analyzes of other rarer tumor types are very interesting but provide limited data as they are only single examples. The analysis of pRCC2 however is very good and provides data across all tumor stages, my main concern here is whether these really are all pRCC2 tumors, as discussed below.

Responses to comments 1, 3, 5, 9, and all minor comments were all satisfactorily.

The response to comment 7 is understandable as pRCC1 tumors are often small, early stage and relatively benign, but not multiple sampling them because of this is rather assuming a result without ever testing it?

Comments 2 and 8 concerned the analysis of samples that were not analyzed by all methodologies or only had single sampling performed. The authors have understandably argued that these samples still provide useful data, which is true, and can be combined with existing data such as the TCGA, that has performed these analyses from single samplings on over 250 papillary tumors. Although, the relatively low number of samples analyzed will have a limited impact on the field. Comment 6 concerned the analysis of single tumor types and the relevance of only analyzing a single tumor. These are relatively rare tumors and the authors understandably have no additional materials, this simply limits the impact of this data as it can not be considered representative of these tumor types.

Comment 4 concerned a fundamental component of this study, which is that pRCC1 and pRCC2 tumors are being compared. I appreciate the authors stating that a single expert opinion was used to define these tumors and I'm sure that this individual has considerable experience but using multiple experts would be beneficial as high grade pRCC1 tumors can be mistaken for pRCC2 tumors. Gain of chromosomes 7 and 17 in combination is very characteristic of pRCC1 and not common in pRCC2 and there are 3 pRCC2 tumors with these chromosomal gains in this study. One pRCC2 also has a VHL mutation and 3p loss that is characteristic of clear cell RCC not pRCC2, did this tumor have a mixed clear cell/pap phenotype? Currently, the study shows no histology images for any tumor analyzed within the study or supplementary data that would allow for others to consider the histologic calls, this would seem essential considering the genetic results shown for the tumors and the purpose of this study.

Reviewer #3 (Remarks to the Author):

I have considered the response to reviewers and revisions to the manuscript and believe they adequately address my prior concerns. My original opinion was that this was an important and well executed study and my criticisms were largely about requests for a bit more insight on a few

points as opposed to corrections. The authors have provided some of these explanations and given reasonable answers for why others cannot be resolved within the scope of the present study. I have no additional criticisms to raise. I still believe this is a nicely executed study, as well as scientifically interesting and valuable for the field.

Reviewer #1 and Reviewer #3 are satisfied with our last round of review comments.

“The authors have satisfactorily addressed all my comments. I now recommend the article for publication.”-- Reviewer #1

“I still believe this is a nicely executed study, as well as scientifically interesting and valuable for the field.”-- Reviewer #3

Review #2 has one comment:

Comment 4 concerned a fundamental component of this study, which is that pRCC1 and pRCC2 tumors are being compared. I appreciate the authors stating that a single expert opinion was used to define these tumors and I'm sure that this individual has considerable experience but using multiple experts would be beneficial as high grade pRCC1 tumors can be mistaken for pRCC2 tumors. Gain of chromosomes 7 and 17 in combination is very characteristic of pRCC1 and not common in pRCC2 and there are 3 pRCC2 tumors with these chromosomal gains in this study. One pRCC2 also has a VHL mutation and 3p loss that is characteristic of clear cell RCC not pRCC2, did this tumor have a mixed clear cell/pap phenotype? Currently, the study shows no histology images for any tumor analyzed within the study or supplementary data that would allow for others to consider the histologic calls, this would seem essential considering the genetic results shown for the tumors and the purpose of this study.

Response: We thank the reviewer for this comment.

The reviewer was concerned that three cases of papillary type 2 carcinomas (pRCC2) had gains of chromosome 7 and 17, which are more frequent in papillary type 1 (pRCC1) than pRCC2. We have asked our pathologist to review these cases again, but he confirmed the diagnosis of pRCC2 for all three cases. We acknowledge the difficulty of distinguishing these subtypes, which often have overlapping features, and so we provided the whole images of these three tumors for the reviewer. These images are accessible at this link:

<https://drive.google.com/drive/folders/1MXgSquFeTBhltJk6jpKrfV2Mu8zla1GS>

The corresponding whole images of these three tumors are included with these names: “pRCC2_1494_04.ndpi”, “pRCC2_1824_13.svs”, and “pRCC2_1568_04.ndpi”.

Moreover, the reviewer asked whether the pRCC2 with a VHL mutation had a mixed clear cell/pap phenotype. In agreement with the reviewer, our pathologist confirmed that this was a pRCC2 with some areas of clear cells, which could explain the VHL mutation. We added this information in the revised manuscript attached to this submission. The corresponding whole image of this tumor is in the Google Drive link, with this name: “pRCC2_1552_03.ndpi”.

The reviewer also asked to view the histological images. We tried multiple times to submit whole images of all tumors to the Editor, so she could send them to the reviewer. Unfortunately, because of the large images size, the Editor was unable to download them. We could not send to the reviewer the link to access the images directly, because it would have required the Editor to reveal the email of the reviewer. We also searched for existing databases where we could

upload the images and provide a link to the reviewer, but the only one we identified (The Cancer Imaging Archive), says on the website that uploading images “usually takes a few months”. Thus, we decided, in agreement with the Editor, to provide to the reviewer the whole images of the tumors he/she was concerned with (the four described above) and provided also close-up images of all tumors analyzed in the manuscript. The folder of the close-up images is named “focused images” in the Google Drive link. These focused images were also included in the revised manuscript as a supplementary file, named “Supplementary Histological Images”. These images will not require a software to view them, so they can be accessed by the general readership with knowledge of histology.

On the revised manuscript, on page 16 (copied also here below), we highlighted in green the changes based on our responses to the comments from reviewer 2.

The study population comprise 39 patients with kidney cancers, including 23 with papillary type 1 (pRCC1); 12 with papillary type 2 (pRCC2); and one each with collecting duct tumor (cdRCC); renal fibrosarcoma rSRC (with negative stain for AE1/AE3, PAX8, CD99, FLI-1, WT1, actine ml, desmine, Myod-1, and HMB45; and positive staining for vimentine and S-100 (focal)); mixed pRCC1/pRCC2 and an unclassified renal cancer with mixed features of pRCC2 and cdRCC (mixRCC). The histological diagnosis was reviewed by an expert uropathologist (S.S.) based on the 2016 World Health Organization (WHO) classification of renal tumors¹. Although our pathologist reviewed all available tissue blocks from each tumor, we cannot exclude the possibility that some of these tumors have mixed histologies (e.g., papillary types 1 and 2) in sections that were not available for histological review. Moreover, the distinction between pRCC1 and pRCC2 can be sometimes murky because of overlapping features and no immunohistochemistry or molecular marker can conclusively distinguish the two subtypes. For example, trisomies 7 and 17 are frequent in pRCC1 but can be also found, less frequently, in pRCC2⁴. There could also be tumors with one dominant histology and a small component of a different histology. For example, pRCC2_1552_03 was a pRCC2 with small areas with clear cells, which may explain the VHL mutation we identified in this tumor. Histological images of all tumors can be found in the Supplementary Histological Images file.

REVIEWERS' COMMENTS:

Reviewer #2 (Remarks to the Author):

I would like to thank the authors for all the efforts that they have gone to and appreciate the addition of text highlighting the difficulty in distinguishing type 1 pRCC and type 2 pRCC and the addition of histologic images to the paper as I feel this could well be very useful. I have no further comments on this study, thank you.

REVIEWERS' COMMENTS:

Reviewer #2 (Remarks to the Author):

I would like to thank the authors for all the efforts that they have gone to and appreciate the addition of text highlighting the difficulty in distinguishing type 1 pRCC and type 2 pRCC and the addition of histologic images to the paper as I feel this could well be very useful. I have no further comments on this study, thank you.

AUTHOR RESPONSE: We appreciate Reviewer's valuable comments, which improve the significance of the manuscript.